# Carbonate $^{206}$Pb / $^{238}$U problems and potential $^{207}$Pb / $^{235}$U fixes

**Pieter Vermeesch[1], Noah McLean[2], Anton Vaks[3], Tzahi Golan[3], Sebastian F. M. Breitenbach[4], and Randall Parrish[5]**

[1]Department of Earth Sciences, University College London, London, WC1E 6BT, United Kingdom

[2]Department of Geology, University of Kansas, Lawrence, KS 66045, United States

[3]Geological Survey of Israel, 9692100 Jerusalem, Israel

[4]Department of Geography and Environmental Sciences, Northumbria University, Newcastle upon Tyne, NE1 8ST, United Kingdom

[5]Department of Geography and Geosciences, University of Portsmouth, Portsmouth, PO1 3QL, United Kingdom TSI

**Correspondence:** Pieter Vermeesch (p.vermeesch@ucl.ac.uk)

**Abstract.** Carbonate U–Pb dating has become a key tool for Quaternary palaeoclimatology and palaeoanthropology beyond the age limit of Th–U disequilibrium dating. U–Pb geochronology is based on the paired radioactive decay of $^{238}$U to $^{206}$Pb and of $^{235}$U to $^{207}$Pb. Current carbonate U–Pb data processing algorithms rely mostly on the $^{206}$Pb / $^{238}$U clock and attach little weight to the $^{207}$Pb / $^{235}$U data. A key weakness of this approach is the need to correct the $^{206}$Pb / $^{238}$U data for initial $^{234}$U / $^{238}$U disequilibrium, which may cause an excess or (occasionally) a deficit in radiogenic $^{206}$Pb compared to secular equilibrium. Uncorrected initial disequilibrium may bias $^{206}$Pb / $^{238}$U dates by up to 4 Myr. We introduce a new disequilibrium correction algorithm, using matrix exponentials. This algorithm can be used to undo the effects of U-series disequilibrium using either an assumed initial composition or a measured set of modern $^{234}$U / $^{238}$U (and optionally $^{230}$Th/$^{238}$U) activity ratios. Using a deterministic Bayesian inversion algorithm, we show that disequilibrium corrections work well for relatively young samples but become unreliable beyond 1.5 Ma and impossible beyond 2 Ma. Using theoretical models and real world examples from Siberia, South Africa, and Israel, we show that the uncertainty of the disequilibrium correction of such old samples exceeds the correction itself. Previous "Monte Carlo" error propagation methods underestimate these uncertainties by up to an order of magnitude. For carbonates older than 2 Ma that likely experienced significant initial $^{234}$U / $^{238}$U disequilibrium, we recommend using the $^{207}$Pb / $^{235}$U isochron method instead of $^{206}$Pb / $^{238}$U geochronology. $^{207}$Pb / $^{235}$U isochrons require only a small and simple correction for initial $^{231}$Pa depletion. This makes $^{207}$Pb / $^{235}$U dating more accurate than $^{206}$Pb / $^{238}$U geochronology. However, the $^{207}$Pb / $^{235}$U method is no panacea. Its precision is limited by the lower abundance of $^{207}$Pb compared to $^{206}$Pb. In some samples, this loss of precision results in a failure to outperform the Bayesian credible intervals of the disequilibrium-corrected $^{206}$Pb / $^{238}$U dates. Such samples remain undatable, unless prior information is available to constrain the initial $^{234}$U / $^{238}$U activity ratios.

## 1 Introduction

Carbonate rocks are only a minor component of the continental crust. However, their scientific importance far outweighs their volumetric abundance. Biogenic carbonates and speleothems document the history of life and of Earth's climate and environment. To generate detailed time series of past changes from these carbonate archives, an accurate and precise chronological framework is essential. This framework is anchored in absolute time using radiometric dating methods, with two techniques commonly employed for this purpose. $^{230}$Th/U dating is the method of choice for young samples whose $^{230}$Th, $^{234}$U, and $^{238}$U activities are out of secular equilibrium (Kaufman and Broecker, 1965; Ludwig, 2003). $^{206}$Pb / $^{238}$U dating is the default method for older rocks, in which the secular equilibrium between $^{230}$Th,

$^{234}$U, and $^{238}$U has been restored (Smith and Farquhar, 1989; Roberts et al., 2020).

Ironically, the absence of detectable $^{234}$U / $^{238}$U disequilibrium compromises the accuracy of the $^{206}$Pb / $^{238}$U
method. Any initial excess or deficit of $^{234}$U and $^{230}$Th affects the $^{206}$Pb / $^{238}$U ratio and, hence, the age estimate derived therefrom. In clean, detritus-free carbonates, it is often safe to assume the absence of initial $^{230}$Th. This assumption is not valid for $^{234}$U, which can be enriched (or occasion-
ally depleted) relative to $^{238}$U by physiochemical processes such as (1) $\alpha$-recoil ejection and preferential leaching of $^{234}$U from $\alpha$-damaged mineral sites and (2) chemical fractionation between preferentially oxidized $^{234}$U$^{6+}$ and $^{238}$U$^{4+}$ (Fleischer, 1982; Porcelli and Swarzenski, 2003).
Corrections for initial $^{234}$U / $^{238}$U disequilibrium can be done either by assuming a specific initial $^{234}$U / $^{238}$U activity ratio or by inferring the initial ratio from any measured residual $^{234}$U / $^{238}$U disequilibrium (Richards et al., 1998; Woodhead et al., 2006; Wendt and Carl, 1985; Engel et al., 2019).
In Sect. 2 of this paper, we review the second approach using matrix exponentials. We show that initial $^{234}$U / $^{238}$U disequilibrium can bias $^{206}$Pb / $^{238}$U dates by up to 4 Myr.

Current disequilibrium correction algorithms use a "Monte Carlo" approach to propagate the errors. In Sect. 3
we will show that this approach can underestimate the analytical uncertainties of $^{206}$Pb / $^{238}$U dates by an order of magnitude when samples are within a few per mil of secular equilibrium (which typically happens before ca. 2 Ma). This observation undermines the results of several published
studies (Vermeesch et al., 2025).

In Sect. 4 we introduce a deterministic Bayesian approach to estimate the uncertainties of disequilibrium-corrected $^{206}$Pb / $^{238}$U dates. We use this alternative algorithm to show that beyond ca. 2 Ma, disequilibrium-corrected $^{206}$Pb / $^{238}$U
dates are impractically imprecise, unless highly enriched initial $^{234}$U / $^{238}$U activity ratios can be ruled out a priori. The large uncertainty associated with the $^{206}$Pb / $^{238}$U method degrades its ability to constrain reliable chronologies for carbonates whose initial disequilibrium has expired. However, a
more accurate approach is available. Following the example of Richards et al. (1998), Neymark and Amelin (2008), Vaks et al. (2020), and others, Sect. 6 makes a case for the little-used $^{207}$Pb / $^{235}$U clock as a replacement for the $^{206}$Pb / $^{238}$U method.
$^{207}$Pb / $^{235}$U isochrons are, essentially, immune to the effects of initial disequilibrium (apart from a minor correction for $^{231}$Pa, which becomes smaller with increasing age). In Sect. 7 we present examples from Siberia and Israel to show that the $^{207}$Pb / $^{235}$U method is more accurate than
the $^{206}$Pb / $^{238}$U method, whilst being less precise for young samples. Both the Bayesian uncertainty estimation method and $^{207}$Pb / $^{235}$U isochrons have been implemented in the IsoplotR toolbox for radiometric geochronology (Sect. 8).

This paper will use the following symbols and notations:

- $n_{38}$, $n_{34}$, $n_{30}$, $n_{26}$ and $n_{06}$: the number of atoms of $^{238}$U, 55 $^{234}$U, $^{230}$Th, $^{226}$Ra and $^{206}$Pb, respectively;

- $\lambda_{38}$, $\lambda_{35}$, $\lambda_{34}$, $\lambda_{32}$, $\lambda_{31}$, $\lambda_{30}$ and $\lambda_{26}$: the radioactive decay constants of $^{238}$U, $^{235}$U, $^{234}$U, $^{232}$Th, $^{231}$Pa, $^{230}$Th and $^{226}$Ra, respectively;

- $[4/8]_t$: the present day $^{234}$U / $^{238}$U activity ratio (i.e. 60 $[\lambda_{34}\,n_{34}]/[\lambda_{38}\,n_{38}]$);

- $[4/8]_i$, $[0/8]_i$, $[1/5]_i$: the initial $^{234}$U / $^{238}$U, $^{230}$Th/$^{238}$U and $^{231}$Pa/$^{235}$U activity ratios, respectively;

- $[4/8]_m$ and $s[4/8]_m$: the measured $^{234}$U / $^{238}$U activity ratio and its standard error, respectively. 65

- $123.456 \pm 0.012$ represents a(n approximate) 95 % confidence interval, whereas 123.456(78) is a more succinct notation that indicates a value of 123.456 with a standard error of 0.078.

## 2  Disequilibrium corrections in a nutshell 70

Even though the U–Pb decay systems consist of numerous steps (14 for the $^{238}$U$\rightarrow^{206}$Pb chain and 11 for the $^{235}$U$\rightarrow^{207}$Pb chain), conventional U–Pb geochronology ignores this complexity, and the method is mathematically treated as a set of simple parent–daughter pairs. This sim- 75
plification is justified once a state of secular equilibrium is established between all the intermediate daughter products in the decay chains. Such secular equilibrium is practically reached after 1 to 2 Myr. As mentioned in Sect. 1, any disequilibrium that might exist prior to this secular equilibrium 80 can be used as a chronometer in its own right.

Initial disequilibrium of the U-decay series affects the accuracy of the U–Pb method. For example, ignoring any initial excess $^{234}$U results in an overestimated $^{206}$Pb / $^{238}$U age, and ignoring any initial $^{231}$Pa deficit results in an underes- 85
timated $^{207}$Pb / $^{235}$U age. Therefore, initial disequilibrium is one mechanism to produce discordant U–Pb results. Extreme $^{234}$U enrichments have been observed in places such as South Africa ($[4/8]_i < 12$; Kronfeld et al., 1994), Siberia ($[4/8]_i < 6$; Vaks et al., 2020) and Japan ($[4/8]_i < 12$; Kurib- 90
ayashi et al., 2025). Using the $[4/8]_i = 12$ value as an upper bound, the maximum effect of initial $^{234}$U / $^{238}$U disequilibrium can be approximated as follows:

$$\Delta(t) \approx \frac{1}{\lambda_{38}} \ln\left[1 + \frac{^{206}\mathrm{Pb}}{^{238}\mathrm{U}} + (12-1)\frac{\lambda_{38}}{\lambda_{34}} + (0-1)\frac{\lambda_{38}}{\lambda_{30}}\right]$$

$$- \frac{1}{\lambda_{38}} \ln\left[1 + \frac{^{206}\mathrm{Pb}}{^{238}\mathrm{U}}\right] \approx \frac{11}{\lambda_{34}} - \frac{1}{\lambda_{30}} = 3.8\,\mathrm{Myr}, \quad (1)$$

where $\lambda_{38} = 0.155125(83)\,\mathrm{Gyr}^{-1}$ (Jaffey et al., 1971), 95 $\lambda_{30} = 9.1705(16)\,\mathrm{Myr}^{-1}$, and $\lambda_{34} = 2.82206(80)\,\mathrm{Myr}^{-1}$ (Cheng et al., 2013). For old carbonates ($> 100$ Ma, say),

a 4 Myr bias may be inconsequential. However, for young carbonates, the relative effect of initial disequilibrium can result in order-of-magnitude levels of bias. A disequilibrium correction is needed to remove this bias.

If the intermediate daughter is sufficiently long lived and the sample is sufficiently young ($t < 5/\lambda$, say) to retain some of its disequilibrium, then the activity ratios can be back-calculated to the time of isotopic closure (assuming subsequent closed-system behaviour). This strategy applies to $^{234}U / ^{238}U$ disequilibrium and, for very young samples, to $^{230}Th / ^{238}U$ disequilibrium. The complex evolution of the U-decay series was first described by Bateman (1908) and subsequently applied to U–Pb geochronology by Ludwig (1977), Wendt and Carl (1985), and Engel et al. (2019). Here we opt for an alternative formulation, using matrix exponentials (Albarède, 1995). For example, the $^{238}U \to ^{206}Pb$ decay chain can be expressed in matrix form as follows:

$$
\frac{\partial}{\partial t}
\begin{bmatrix} n_{38} \\ n_{34} \\ n_{30} \\ n_{26} \\ n_{06} \end{bmatrix}
=
\begin{bmatrix}
-\lambda_{38} & 0 & 0 & 0 & 0 \\
\lambda_{38} & -\lambda_{34} & 0 & 0 & 0 \\
0 & \lambda_{34} & -\lambda_{30} & 0 & 0 \\
0 & 0 & \lambda_{30} & -\lambda_{26} & 0 \\
0 & 0 & 0 & \lambda_{26} & 0
\end{bmatrix}
\begin{bmatrix} n_{38} \\ n_{34} \\ n_{30} \\ n_{26} \\ n_{06} \end{bmatrix},
\quad (2)
$$

where $\lambda_{26} = 0.4332(19)\,\mathrm{kyr}^{-1}$ (Audi et al., 2003), and the shortest-lived intermediate daughters ($< 1\,\mathrm{kyr}$ half-lives) have been omitted. The solution to Eq. (2) is a so-called matrix exponential:

$$
\begin{bmatrix} n_{38} \\ n_{34} \\ n_{30} \\ n_{26} \\ n_{06} \end{bmatrix}
= \mathrm{expm}
\left(
\begin{bmatrix}
-\lambda_{38} & 0 & 0 & 0 & 0 \\
\lambda_{38} & -\lambda_{34} & 0 & 0 & 0 \\
0 & \lambda_{34} & -\lambda_{30} & 0 & 0 \\
0 & 0 & \lambda_{30} & -\lambda_{26} & 0 \\
0 & 0 & 0 & \lambda_{26} & 0
\end{bmatrix} t
\right)
\begin{bmatrix} n_{38} \\ n_{34} \\ n_{30} \\ n_{26} \\ n_{06} \end{bmatrix}_{\mathrm{i}},
\quad (3)
$$

which expresses the present-day amounts of the different isotopes as a function of the initial amounts. An interesting result is obtained by setting $t = \infty$ in Eq. (3) to estimate the activity ratio under secular equilibrium:

$$
[4/8]_\infty = \frac{\lambda_{234}}{\lambda_{234} - \lambda_{238}} = 1.000055.
\quad (4)
$$

Note that this activity ratio is *not* exactly equal to unity. This is because 0.0055 % of $^{238}U$ is lost during $^{234}U$'s mean lifetime of $1/\lambda_{34} = 354\,\mathrm{kyr}$. Equation (3) can also be inverted to express the initial amounts as a function of the present-day amounts:

$$
\begin{bmatrix} n_{38} \\ n_{34} \\ n_{30} \\ n_{26} \\ n_{06} \end{bmatrix}_{\mathrm{i}}
= \mathrm{expm}
\left(
-\begin{bmatrix}
-\lambda_{38} & 0 & 0 & 0 & 0 \\
\lambda_{38} & -\lambda_{34} & 0 & 0 & 0 \\
0 & \lambda_{34} & -\lambda_{30} & 0 & 0 \\
0 & 0 & \lambda_{30} & -\lambda_{26} & 0 \\
0 & 0 & 0 & \lambda_{26} & 0
\end{bmatrix} t
\right)
\begin{bmatrix} n_{38} \\ n_{34} \\ n_{30} \\ n_{26} \\ n_{06} \end{bmatrix}.
\quad (5)
$$

Equations (3) and (5) can be used to construct a concordia diagram in the presence of disequilibrium. If measured activity ratios are used to infer the initial conditions, then the concordia line terminates where those inferred activity ratios reach unrealistic values (e.g. $[4/8]_{\mathrm{i}} = 500$ and $[0/8]_{\mathrm{i}} = 0$; Fig. 1a). Beyond 10 or so $^{234}U$ half-lives, it becomes very difficult to estimate $[4/8]_{\mathrm{i}}$ from $[4/8]_{\mathrm{m}}$, and it is even more

difficult to quantify the analytical uncertainty of the disequilibrium correction. In Sect. 3 we review the current "Monte Carlo" approach to uncertainty estimation for initial disequilibrium correction, and in Sect. 4 we propose an alternative "Bayesian" approach, which offers significant advantages for samples that are close to secular equilibrium.

## 3 "Monte Carlo" uncertainty estimation

Existing data processing software for disequilibrium-corrected $^{206}Pb / ^{238}U$ geochronology, such as DQPB (Pollard et al., 2023), estimates the uncertainty of the disequilibrium correction by Monte Carlo simulation. Given a linear array of isotopic data in Tera–Wasserburg space (i.e. $n_{07}/n_{06}$ vs. $n_{38}/n_{06}$), paired with an activity ratio measurement $[4/8]_{\mathrm{m}}$ with standard error $s[4/8]_{\mathrm{m}}$, this approach works as follows:

1. Draw a random value $[4/8]_t$ from a normal distribution with mean $[4/8]_{\mathrm{m}}$ and standard deviation $s[4/8]_{\mathrm{m}}$.

2. Fit a straight line to the U–Pb data and find the $[4/8]_{\mathrm{i}}$ value and isochron age ($t$) that are consistent with both the U–Pb measurements and $[4/8]_t$. In other words, use Eq. (5) to estimate $[4/8]_{\mathrm{i}}$ from $[4/8]_t$, and repeat this for different values of $t$ until the linear fit to the U–Pb data is optimized.

3. Repeat steps (1) and (2) until the entire distribution of $[4/8]_t$ values has been sampled.

4. If step (2) fails or produces physically impossible results (e.g. $t < 0$), then ignore the corresponding $[4/8]_t$ value. Otherwise add the $[4/8]_{\mathrm{i}}$ and $t$ values to a list of acceptable results.

5. Use the spread of the acceptable $[4/8]_{\mathrm{i}}$ and $t$ values to quantify their respective uncertainties.

For the purpose of the present study, we have implemented our own version of this algorithm (Vermeesch, 2025) [TS3], using R and IsoplotR (Vermeesch, 2018). The only major difference between our code and DQPB is that it does not sample the $[4/8]_{\mathrm{m}}$ distribution randomly but uses a targeted approach to sample $[4/8]_{\mathrm{m}}$ as a sequence of regularly spaced normal quantiles. This is faster and produces deterministic results that do not depend on the seed of a random number generator. Figure 1 summarizes the application of this approach using the "Corchia" dataset of Pollard et al. (2023), producing identical results to DQPB. To reflect this equivalence of outcomes, we will refer to our version of the algorithm as a "Monte Carlo" method, despite the fact that it does not actually use a random number generator.

Next, let us apply the same approach to older materials such as sample AV03 (Bolt's Farm, South Africa) of Pickering et al. (2019). The uncorrected U–Pb isochron age for this

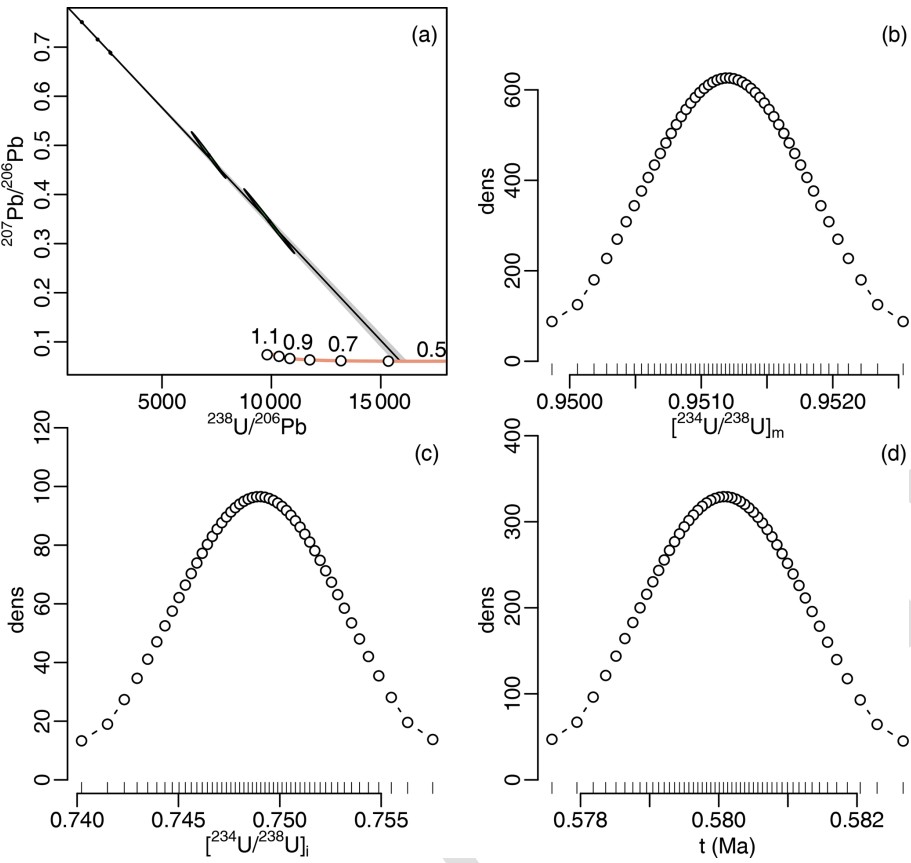

**Figure 1.** Output of the "Monte Carlo" algorithm for the Corchia dataset of Pollard et al. (2023). **(a)** Tera–Wasserburg concordia diagram with disequilibrium-corrected isochron ($t = 0.5804 \pm 0.0086$ Ma); **(b)** 50 representative samples from the distribution of $^{234}$U / $^{238}$U activity ratio measurements; **(c)** the corresponding initial $^{234}$U / $^{238}$U activity ratios; and **(d)** the isochron ages corresponding to the $[4/8]_i$ values presented in panel **(c)**. TS2

sample is $5.6 \pm 0.9$ Ma, which is 22 half-lives of $^{234}$U. Consequently, the measured present-day $^{234}$U / $^{238}$U activity ratio is statistically indistinguishable from secular equilibrium, at $[4/8]_m = 1.0046 \pm 0.0063$. Despite the lack of measurable disequilibrium, the "Monte Carlo" approach appears to have successfully applied a disequilibrium correction, resulting in a corrected age that is less than half the uncorrected age, with a precision of better than 12 % (Fig. 2). How is this possible? The answer lies in the rejected solutions (step 4 of the algorithm), which are marked in black in Fig. 2b. Ignoring these "physically impossible" initial ratios suppresses the equilibrium solutions and skews the distribution of Monte Carlo solutions towards high $[4/8]_i$ values (Fig. 2c) and young ages (Fig. 2d).

To demonstrate that the result of Fig. (2) is wrong, let us replace the measured $^{234}$U / $^{238}$U activity ratio with the equilibrium ratio (Eq. 4):

$$[4/8]_m = [4/8]_\infty \pm 0.0063.$$

Plugging this value into the "Monte Carlo" algorithm yields an impossible result (Fig. 3). It has applied a disequi-

librium correction without any actual disequilibrium, by ignoring exactly half of the $[4/8]_t$ distribution (Fig. 3b). This was necessary because, for this old sample, essentially any $[4/8]_t$ value that is less than the equilibrium ratio would require a negative $[4/8]_i$ ratio or a negative isochron age $t$.

## 4 A Bayesian approach

The previous section showed that the "Monte Carlo" algorithm produces incorrect results for samples whose $[4/8]_t$-distributions fall within, say, 3 standard errors (i.e. $3 \times s[4/8]_m$) from secular equilibrium. One way to address this issue is for the "Monte Carlo" algorithm to issue a warning when the $[4/8]_m$ value is close to $[4/8]_\infty$. This is the approach taken by DQPB (Pollard et al., 2023). In this section we introduce an alternative approach that automatically handles problematic cases, without the need to define a nominal "applicability cutoff". Given any values of $[4/8]_m$ and $s[4/8]_m$, our new "Bayesian" algorithm proceeds as follows:

1. Define a prior distribution for $[4/8]_i$. In a first instance, we will use a uniform distribution that stretches between

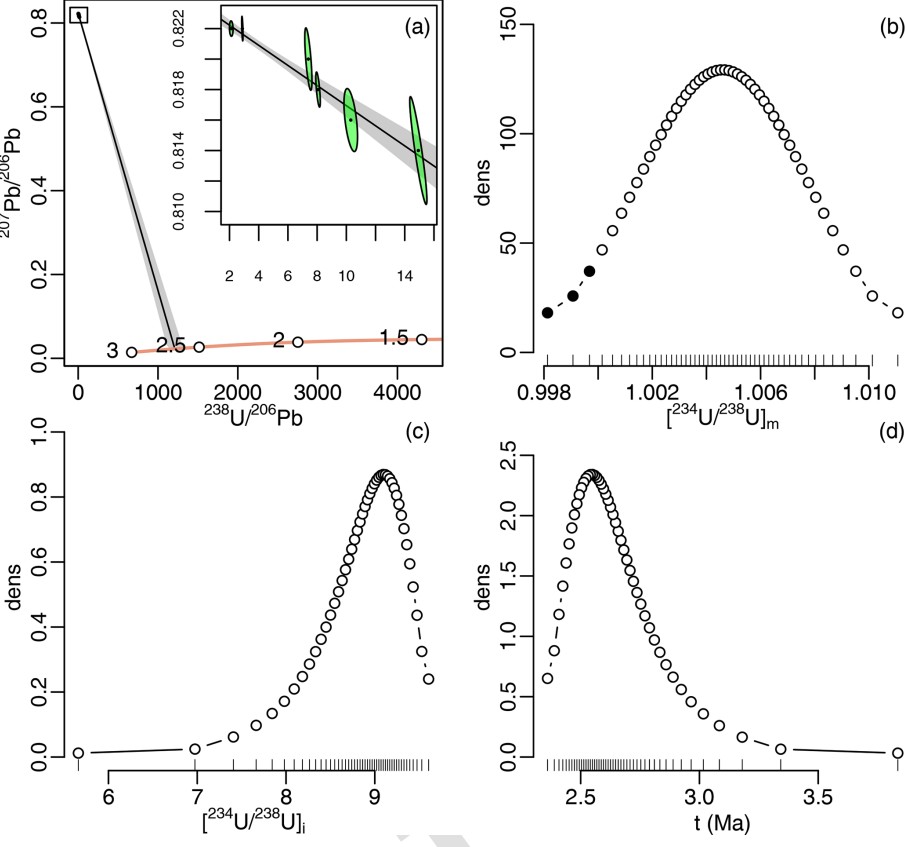

**Figure 2.** Output of the "Monte Carlo" algorithm for sample AV03 of Pickering et al. (2019). Panels **(a)**–**(d)** are as in Fig. 1. The black dots in panel **(b)** mark synthetic replicates that are rejected because they yield physically impossible $[4/8]_i$ and/or $t$ values. The results shown in panels **(c)** and **(d)** are consistent with the published values.

a minimum $[4/8]_i$ value $m = 0$ and a maximum $[4/8]_i$ value $M = 20$. However, this uniform distribution can easily be replaced by a more informative prior. One flexible way to capture a diversity of prior information is the logistic normal distribution:TS4

$$\ln\left(\frac{[4/8]_i - m}{M - m}\right) \sim \mathcal{N}(\mu, \sigma^2), \tag{6}$$

where $\mu$ and $\sigma$ are the location and dispersion parameters of the distribution, respectively. An application of this informative prior will be given in Sect. 7.

2. Draw a random sample from the prior distribution, carry out a constrained isochron regression, and register the resulting age ($t$) and corresponding $[4/8]_t$ value. In other words, use Eq. (3) to estimate $[4/8]_t$ from $[4/8]_i$, and repeat this for different values of $t$ until the linear fit to the U–Pb data is optimized. Register the likelihood of this linear fit using the same algorithm as used for regular U–Pb isochron regression (Ludwig, 1998; Vermeesch, 2020).

3. Calculate the likelihood of the inferred $[4/8]_t$ values under a normal distribution with mean $[4/8]_m$ and stan-

dard deviation $s[4/8]_m$. Combine with the likelihood of the linear fit (obtained in step 2) to produce the "posterior" probability of initial ratios.

4. Repeat steps (2) and (3) to constrain the posterior distributions of $[4/8]_i$ and $t$. This can be done either using a Markov chain or with a targeted approach of appropriately spaced $[4/8]_i$ values.

Applying this method to the 580 ka Corchia example (Fig. 4) yields essentially identical results to the "Monte Carlo" algorithm (Fig. 1).

However, when the Bayesian approach is applied to the older Bolt's Farm data (Fig. 5), it produces a very different result than the "Monte Carlo" approach (Fig. 2). The posterior distributions for Bolt's Farm sample AV03 (shown in Fig. 5c and d) still have maxima at $[4/8]_i = 9$ and $t = 2.6$ Ma, just like the "Monte Carlo" distributions (Fig. 2c and d). But unlike the "Monte Carlo" solution, the result of the Bayesian approach also assigns a significant probability to older ages, including the uncorrected U–Pb date of 5.6 Ma. The similarity of this posterior distribution to the prior distribution reflects the fact that the measured $^{234}U / ^{238}U$ activity ratio contains relatively little information. The resulting uncertain-

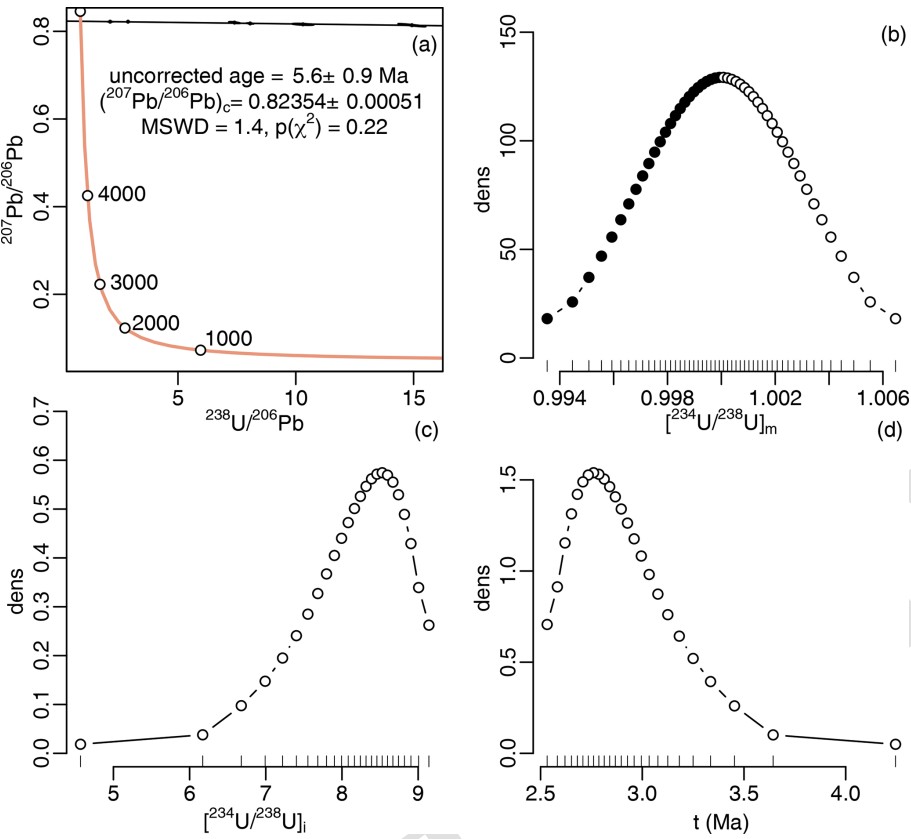

**Figure 3.** The same data as Fig. 2 but replacing $[4/8]_m$ with the equilibrium ratio. Note that half of the synthetic replicates have been rejected (black circles). Even though there is absolutely no evidence for disequilibrium, the "Monte Carlo" produces a corrected isochron age **(d)** that is half the uncorrected value **(a)**. This result is clearly wrong.

ties are large but correctly reflect our ignorance about the true extent of the disequilibrium in this case.

Finally, changing the $[4/8]_m$ ratio to the equilibrium value (Fig. 5) produces a posterior distribution that is nearly iden-
5 tical to the prior distribution. This means that the likelihood function contains almost no information. In other words, the measured $^{234}U / ^{238}U$ activity ratio does not tell us anything about the initial disequilibrium (except that $[4/8]_i < 10$). If it cannot be ruled out that the sample may have experienced
extreme $^{234}U / ^{238}U$ disequilibrium, then it is not possible to undo the effects of this disequilibrium using the modern (measured) $^{234}U / ^{238}U$ activity ratio.

## 5   The case against $^{206}Pb / ^{238}U$ dating of old carbonates

The applicability range of the $^{206}Pb / ^{238}U$ method depends on the $[4/8]_i$ ratio and on the precision of the $[4/8]_m$ measurements. Although sub-per-mil level analytical uncertainties can be routinely achieved for individual $[4/8]_m$ measurements, the external reproducibility is likely worse than this in
most samples. This is due to a combination of two competing factors. First, $[4/8]_i$ is negatively correlated with U concen-

tration (Osmond and Cowart, 1976; Zhou et al., 2005; Kurib-ayashi et al., 2025). Second, the U concentration must exhibit large variations to form a statistically robust $^{206}Pb / ^{238}U$ isochron.                                                                                                    25

The combination of these two effects has the potential to cause intra-sample variations in $[4/8]_m$, exceeding the analytical uncertainties. The exact magnitude of the dispersion is unknown because most speleothem U–Pb dating studies report only one or a few $[4/8]_m$ values per isochron. Here we   30 will assume a conservative value of $s[4/8]_m = 2‰$, based on a collection of 12 speleothems analysed by Walker et al. (2006).

Disequilibrium corrections using measured $^{234}U / ^{238}U$ activity ratios are only feasible if those activity ratios are sta-   35 tistically distinguishable from secular equilibrium. To turn these conclusions into quantitative guidelines, let us define "statistically distinguishable" as "at least $3 \times s[4/8]_m$ re-moved from secular equilibrium". Using this definition, a sample with $[4/8]_i < 2.7$ would become indistinguishable   40 from secular equilibrium after ca. 2 Ma. In other words, it is impossible to correct a 2 Ma sample whose $[4/8]_i = 2.7$, say. The uncorrected $^{206}Pb / ^{238}U$ isochron age of such a sample would be 2.6 Ma, corresponding to a bias of 30 %. Table 1

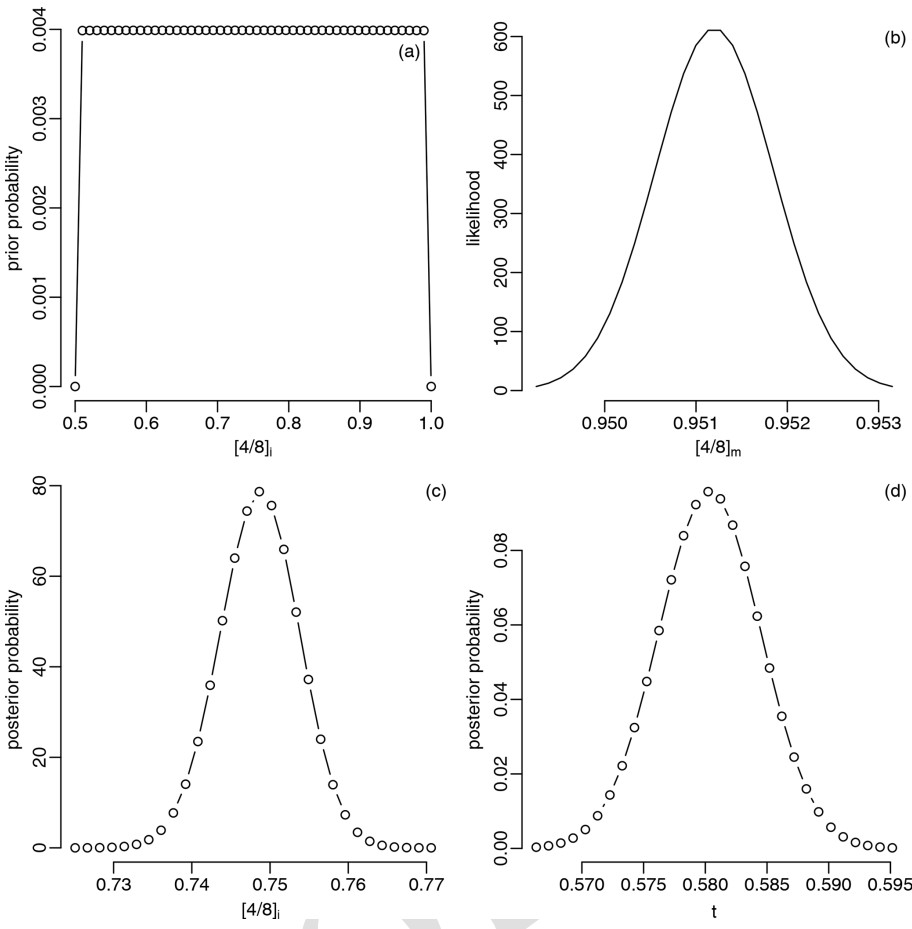

**Figure 4.** The same data as Fig. 1 but using the Bayesian approach. *y* axes display the prior probability **(a)**, likelihood **(b)**, and posterior probabilities **(c, d)**, respectively. For this relatively young sample, the Bayesian method yields similar results to the "Monte Carlo" solution.

**Table 1.** Sensitivity test of the $^{206}$Pb / $^{238}$U method using selected ages.

| True age (Ma) | 0.5 | 1 | 1.5 | 2 | 2.5 | 3.0 |
|---|---|---|---|---|---|---|
| Min. resolvable $[4/8]_i$ | 1.02 | 1.10 | 1.41 | 2.68 | 7.89 | 29.2 |
| Max. bias (%) | 1.1 | 3.2 | 9.5 | 30 | 97 | 333 |

shows the outcomes of the same exercise for a range of other ages.

Figure 6 presents a more extensive exploration of the magnitude (panel a) and precision (panel b) of disequilibrium-corrected $^{206}$Pb / $^{238}$U geochronology assuming the aforementioned 2‰ reproducibility. Alternative versions of this diagram can be generated by modifying the reproducibility value in the R code provided in the Supplement of Vermeesch (2025).

Based on these considerations, we judge carbonate $^{206}$Pb / $^{238}$U geochronology to be unreliable beyond ca. 1.5 Ma and impossible beyond ca. 2 Ma unless initial $^{234}$U / $^{238}$U disequilibrium can be confidently ruled out (Fig. 6). However, there is a solution to the conundrum of $^{234}$U / $^{238}$U disequilibrium. This solution is the $^{207}$Pb / $^{235}$U isochron method (Richards et al., 1998; Engel et al., 2019; Vaks et al., 2020; Vermeesch et al., 2025).

## 6 A potential $^{207}$Pb / $^{235}$U fix to $^{206}$Pb / $^{238}$U's problems

In the previous section, we showed that the $^{206}$Pb / $^{238}$U method's accuracy is hampered by the extreme enrichment (up to double the equilibrium value or more) of $^{234}$U observed in certain groundwaters (e.g. Osmond and Cowart, 1976; Kronfeld et al., 1994; Kuribayashi et al., 2025). This problem can be solved by avoiding $^{234}$U altogether and sidestepping the $^{238}$U–$^{206}$Pb decay chain in favour of the $^{235}$U–$^{207}$Pb decay chain (Neymark and Amelin, 2008).

There are two kinds of $^{207}$Pb / $^{235}$U isochrons. The simplest kind plots $^{204}$Pb / $^{207}$Pb ratios against $^{204}$Pb / $^{235}$U ratios, defining the following linear relationship:

$$\left[\frac{^{204}\text{Pb}}{^{207}\text{Pb}}\right] = \left[\frac{^{204}\text{Pb}}{^{207}\text{Pb}}\right]_i \left\{1 - \left[\frac{^{235}\text{U}}{^{207}\text{Pb}}\right](\exp[\lambda_{35}t] - 1)\right\}, \quad (7)$$

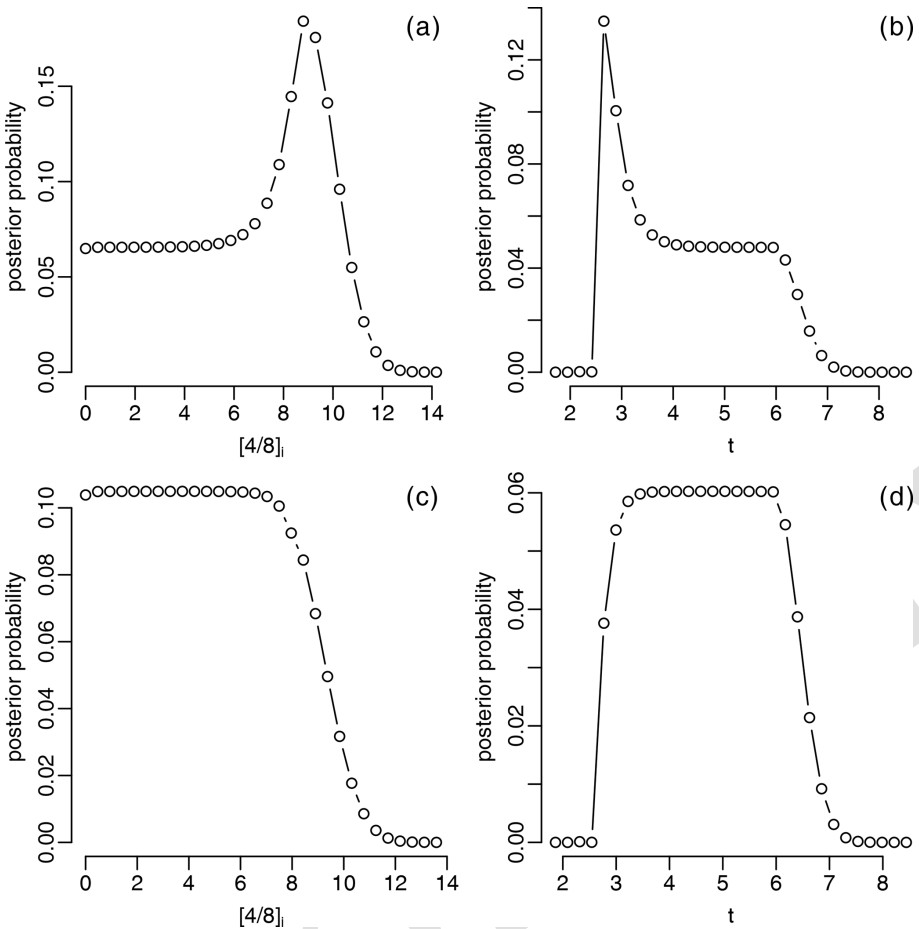

**Figure 5.** Posterior distributions of $[4/8]_i$ and $t$ for sample AV03 of Pickering et al. (2019). Panels **(a)** and **(b)** represent the original U–Pb data of Fig. 2, whereas panels **(c)** and **(d)** show the modified data of Fig. 3. Likelihood functions are provided in panel **(b)** of the latter two figures. A uniform prior was used but is not shown. The modes of the posterior distributions agree with the modes of the "Monte Carlo" solutions. However, whereas the "Monte Carlo" algorithm suggests a high degree of confidence in the disequilibrium correction, the Bayesian approach shows that one cannot rule out a much higher age of the sample, including the uncorrected date of 5.6 Ma.

where $^{204}$Pb is used as a proxy for common Pb. Alternatively, one can also use $^{208}$Pb to fulfil this role. This gives rise to a $^{208}$Pb$_i$/$^{207}$Pb vs. $^{235}$U / $^{207}$Pb isochron, where $^{208}$Pb$_i$ is the non-radiogenic $^{208}$Pb component (with the decay products of $^{232}$Th removed). Equation (7) is then replaced with

$$\left[\frac{^{208}\text{Pb}_i}{^{207}\text{Pb}}\right] = \left[\frac{^{208}\text{Pb}}{^{207}\text{Pb}}\right]_i \left\{1 - \left[\frac{^{235}\text{U}}{^{207}\text{Pb}}\right](\exp[\lambda_{35}t] - 1)\right\}, \quad (8)$$

where

$$\left[\frac{^{208}\text{Pb}_i}{^{207}\text{Pb}}\right] = \left[\frac{^{208}\text{Pb}}{^{207}\text{Pb}}\right] - \left[\frac{^{232}\text{Th}}{^{207}\text{Pb}}\right]\exp[\lambda_{32}t] + 1, \quad (9)$$

in which $\lambda_{32} = 0.0495(25)\,\text{Gyr}^{-1}$ (Le Roux and Glendenin, 1963) and the $^{232}$Th / $^{207}$Pb ratio can be obtained from the product of the $^{232}$Th / $^{238}$U, $^{238}$U / $^{235}$U, and $^{232}$Th / $^{207}$Pb ratios. Because Th is insoluble in water, radiogenic $^{208}$Pb is often absent from carbonates. Therefore, it is generally safe to assume that $\left[^{208}\text{Pb}_i/^{207}\text{Pb}\right] \approx \left[^{208}\text{Pb}/^{207}\text{Pb}\right]$.

The $^{235}$U → $^{207}$Pb method has just one long-lived intermediate daughter, $^{231}$Pa, which requires a correction. Due to $^{231}$Pa's short half-life of 32.65 kyr ($\lambda_{31} = 21.158(71)\,\text{Myr}^{-1}$; Audi et al., 2003), it is generally not possible to measure any remaining disequilibrium in the Myr time range, where the $^{235}$U → $^{207}$Pb method offers a tangible advantage over the $^{238}$U → $^{206}$Pb method. Therefore, the $[1/5]_i$ value must be assumed.

Pa is chemically similar to Th and insoluble in water. Therefore, $^{231}$Pa is always depleted relative to $^{235}$U in carbonates, so whereas $[4/8]_i$ can vary anywhere between 0 and 12 (Osmond and Cowart, 1976), $[1/5]_i$ is always less than 1 and can be safely assumed to be zero. In a worst-case scenario, in which one assumes $[1/5]_i = 0$ but the true activity ratio is $[1/5]_i = 1$, this would only bias the $^{235}$Pb / $^{235}$U age by a relatively small amount (Table 2).

The degree of potential bias of the $^{207}$Pb / $^{235}$U method decreases with increasing age, unlike the $^{206}$Pb / $^{238}$U method,

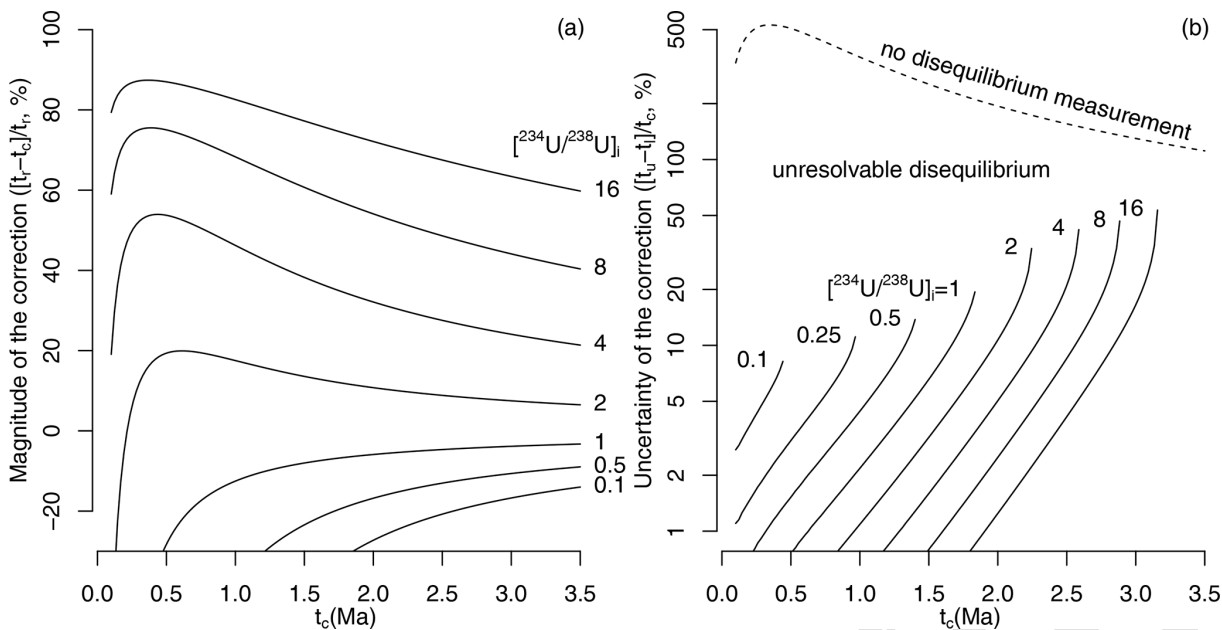

**Figure 6.** Nomogram to assess the applicability of the $^{206}Pb / ^{238}U$ method in the presence of different degrees of initial $^{234}U / ^{238}U$ disequilibrium. Panels **(a)** and **(b)** visualize CE1 the magnitude and the precision of the disequilibrium correction, respectively, for selected values of the initial activity ratios $[4/8]_i$. $t_r$ is the uncorrected date, assuming secular equilibrium. $t_c$ is the true age, which equals the disequilibrium-corrected date using the expected $[4/8]_m$ value. $t_u$ and $t_l$ are the disequilibrium-corrected dates using present-day $^{234}U / ^{238}U$ activity ratios of $[4/8]_m + 2‰$ and $[4/8]_m − 2‰$, respectively. The dashed line in panel **(b)** marks the relative uncertainty interval when no disequilibrium measurement is available, defined as the difference between corrected dates assuming initial $^{234}U / ^{238}U$ activity ratios of 1 and 12.

**Table 2.** Sensitivity test of the $^{207}Pb / ^{235}U$ method against $^{231}Pa$ disequilibrium.

| True age (Ma) | 0.5 | 1 | 1.5 | 2 | 2.5 | 3.0 |
|---|---|---|---|---|---|---|
| Maximum bias (%) | 9.4 | 4.7 | 3.1 | 2.4 | 1.9 | 1.6 |

whose bias increases with age (Table 1). In this sense, the $^{207}Pb / ^{235}U$ and $^{206}Pb / ^{238}U$ methods are complementary to each other. The $^{207}Pb / ^{235}U$ is most accurate for samples older than 1 Ma, whereas the $^{206}Pb / ^{238}U$ is more accurate for samples younger than 1 Ma. Note that the latter is similar to the applicability range of the $^{230}Th / U$ method, so one could argue that disequilibrium-corrected $^{206}Pb / ^{238}U$ dating is of limited use to carbonate U–Pb geochronology (except to infer $[4/8]_i$; Engel et al., 2019). Although the $^{207}Pb / ^{235}U$ method outperforms the $^{206}Pb / ^{238}U$ method at ca. 1 Ma in terms of accuracy, its poorer precision means that its potential benefits do not materialize until ca. 2 Ma. In the next section, we will demonstrate this by applying both methods to three different case studies.

## 7 Case studies

Having made a largely theoretical case against $^{206}Pb / ^{238}U$ dating and for $^{207}Pb / ^{235}U$ dating of old carbonates that are suspected to have experienced initial $^{234}U / ^{238}U$ disequilibrium, we will now compare and contrast the two chronometers using three practical case studies. The first example will demonstrate the accuracy of the $^{207}Pb / ^{235}U$ method by showing its consistency with disequilibrium-corrected $^{206}Pb / ^{238}U$ dates of young ($< 2$ Ma) samples.

The second example uses ID-TIMS data to serve two purposes. First, it will show that the $^{207}Pb / ^{235}U$ method produces more accurate, more consistent, and more precise results than the $^{206}Pb / ^{238}U$ method for older ($> 2$ Ma) carbonates. Second, it will demonstrate how the Bayesian framework can use prior information to overcome the inaccuracy of the $^{206}Pb / ^{238}U$ method.

In the third case study we apply the $^{207}Pb / ^{235}U$ method to LA-ICP-MS data using $^{208}Pb$ as a proxy for common Pb. In addition to highlighting a successful application where the $^{207}Pb / ^{235}U$ method produces demonstrably superior results to the $^{206}Pb / ^{238}U$ method, this dataset also illustrates limitations of the $^{207}Pb / ^{235}U$ approach with a sample that yields a precise $^{206}Pb / ^{238}U$ isochron and an unusably imprecise $^{207}Pb / ^{235}U$ isochron.

### 7.1 ID-ICP-MS data from Siberia

A rich dataset of 72 speleothem dates is available from the Botovskaya and Ledyanaya Lenskaya (LLC) caves in

Siberia. Vaks et al. (2020) used the U–Pb method to extend an important palaeoclimatological archive from these caves that was previously dated using the $^{230}$Th–U disequilibrium method (Vaks et al., 2013a). Samples were analysed by isotope dilution ICP-MS and were found to exhibit a significant level of $^{234}$U / $^{238}$U disequilibrium. U–Pb ages were estimated using a two-step procedure. First, the common-Pb contribution was removed by two-point isochron regression through an inherited composition that was inferred by inspection of apparent linear trends in Tera–Wasserburg concordia space. Second, a $^{234}$U / $^{238}$U disequilibrium correction was applied to the radiogenic end-member composition, using the procedures described in Sect. 2. This correction combined the measured $^{234}$U / $^{238}$U activity ratios with an assumed absence of initial $^{230}$Th and $^{231}$Pa.

The inferred $[4/8]_i$ values were ca. 2 and 3–5 for LLC and Botovskaya cave, respectively. This corresponds to an age correction of 15 % for LLC and 60 % for Botovskaya cave (Fig. 7). Uncertainties were estimated using the Bayesian procedure of Sect. 4, making the optimistic assumption that the analytical uncertainty of the $[4/8]_m$ measurements faithfully captures all sources of dispersion. The scatter of the $[4/8]_i$ values suggests that this may not be the case. This caveat notwithstanding, the disequilibrium-corrected $^{206}$Pb / $^{238}$U and $^{207}$Pb / $^{235}$U ages overlap within uncertainty in all but four of the samples.

The $^{207}$Pb / $^{235}$U age uncertainty is invariably larger than the $^{206}$Pb / $^{238}$U age uncertainty. In fact, below ca. 1 Ma, it could be argued that the $^{207}$Pb / $^{235}$U age uncertainties are unusably imprecise ($s[t]/t > 50$ %). However, above ca. 1 Ma, the uncertainty reduces to acceptable levels ($s[t]/t < 5$ %). Extrapolating this trend further into the past confirms the earlier assertion that beyond ca. 2 Ma, the $^{207}$Pb / $^{235}$U method outperforms the $^{206}$Pb / $^{238}$U method in both accuracy and precision.

## 7.2   ID-TIMS data for ASH-15

ASH-15 is a carbonate U–Pb dating reference material sourced from a flowstone in Ashalim cave of southern Israel (Nuriel et al., 2021). 37 ID-TIMS measurements were obtained from the flowstone, including 12 from horizon D and 25 from horizon K. The latter are shown in Fig. 8. Nuriel et al. (2021) report an uncorrected semitotal-Pb / U isochron age of $2.965 \pm 0.011$ Ma for ASH-15.

No $^{234}$U / $^{238}$U activity ratio measurements are available for ASH-15D and ASH-15K. However, two other horizons of the same flowstone (ASH-15A+B and ASH-15C1) are characterized by $[4/8]_m$ values of $0.99939 \pm 0.00108$ and $0.99925 \pm 0.0015$, indistinguishable from secular equilibrium. Younger flowstones in Ashalim cave yield an average $[4/8]_i$ value of 1.0470, with a standard deviation of 0.01492 (Vaks et al., 2010, 2013b). A wider survey of 904 speleothem samples dated in southern and central Israel by Chaldekas et al. (2022) have average $[4/8]_i$ values of $1.081 \pm 0.138$.

Despite this lack of observable disequilibrium, Mason et al. (2013) suggest a $[4/8]_i$ value of 1.5–2.0 to explain the minor degree of discordance of the common-Pb-corrected Tera–Wasserburg ratios.

To investigate the effect of initial U-series disequilibrium on ASH-15, TS5 Fig. 8a and b apply the Bayesian inversion algorithm to the ASH-15K data, using the $[4/8]_m$ value of ASH-15C1 and assuming that $[0/8]_i = 0$ (i.e. no initial $^{230}$Th). In a first attempt, we will use the same uniform prior from $m = 0$ to $M = 20$ as before. This results in a disequilibrium-corrected $^{206}$Pb / $^{238}$U-isochron age of $3.47 + 0.053/ - 0.794$ Ma and an inferred $[4/8]_i$ ratio of $0.067 + 2.23/ - 0.052$. As expected, initial equilibrium is very plausible with $P([4/8]_i > 1.0) = 0.47$. The initial activity ratio preferred by Mason et al. (2013) cannot be ruled out either but is less likely, with $P([4/8]_i > 1.5) = 0.25$ and $P([4/8]_i > 2.0) = 0.09$.

In a second attempt, we used the $[4/8]_i$ values of Chaldekas et al. (2022) to construct an informative prior, using the logistic normal formulation of Eq. (6) with $m = 1$, $M = 3$, $\mu = 1.081$, and $\sigma = 0.2$. This produces a posterior distribution for $[4/8]_i$ that is, essentially, identical to the prior, confirming that the $[4/8]_m$ data carry virtually no additional information. The corresponding disequilibrium-corrected $^{206}$Pb / $^{238}$U-isochron age is $3.107 \pm 0.065$ Ma.

In contrast with the widely varying scenarios for the $^{206}$Pb / $^{238}$U method, the $^{207}$Pb / $^{235}$U isochron age calculation is straightforward:

1. an uncorrected $^{207}$Pb / $^{235}$U isochron age of $3.039 \pm 0.068$ Ma, assuming $[1/5]_i \approx 1$,

2. a corrected $^{207}$Pb / $^{235}$U isochron age of $3.086 \pm 0.068$ Ma, assuming $[1/5]_i = 0$.

These are nearly identical to the $^{206}$Pb / $^{238}$U date using the informative $[4/8]_i$ prior. We would like to conclude the discussion of ASH-15 by remarking that disequilibrium issues do not affect the suitability of this sample as a reference material for in situ U–Pb geochronology. This is because standardization is done relative to the *uncorrected* isotopic composition (Horstwood et al., 2016).

## 7.3   LA-ICP-MS data from Siberia

For the final case study, we return from Israel to the Botovskaya cave deposits in Siberia. Section 7.1 and Fig. 7 show abundant evidence that this cave is strongly enriched in initial $^{234}$U, with $[4/8]_i$ values ranging from 3 to 5 according to the results of Vaks et al. (2020). The effect of this strong disequilibrium can be confidently undone for the young ($< 500$ ka) speleothems shown in Fig. 7. However, Botovskaya cave also contains speleothems that are considerably older than this, going all the way back to the Plio-Pleistocene. By now it should be clear that the $^{206}$Pb / $^{238}$U method is ill suited to unlock this older archive. Figure 9

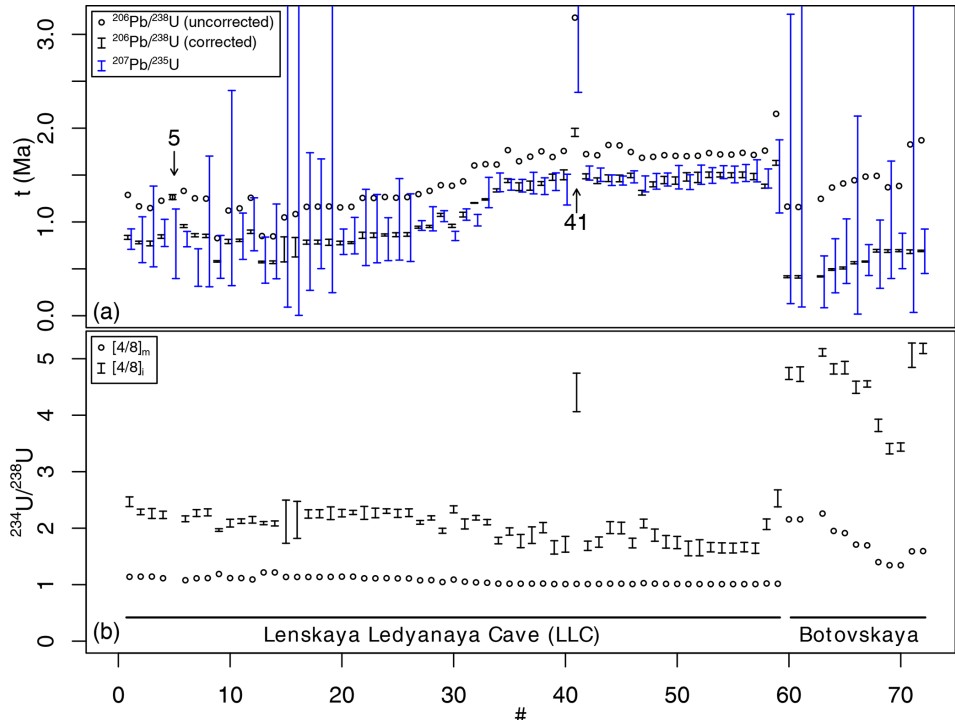

**Figure 7.** Reanalysis of the speleothem data of Vaks et al. (2020). This record stacks together several speleothems, which are arranged in stratigraphic order for each cave. **(a)** Uncorrected (circles) and corrected (black error bars) $^{206}$Pb / $^{238}$U dates, juxtaposed next to the $^{207}$Pb / $^{235}$U dates (blue error bars) for the same samples. **(b)** Measured ($[4/8]_m$, circles) and inferred initial ($[4/8]_i$, error bars) $^{234}$U / $^{238}$U activity ratios. All error bars represent Bayesian 95 % credible intervals. Sample 5 does not have a $[4/8]_m$ measurement and was assumed to be in secular equilibrium. Sample 41 is an outlier that has an anomalously high common-Pb concentration and is only included for the sake of completeness.

summarizes some preliminary U–Pb results from two of these older cave deposits (sample SB-1625-22 and sample SB-72-8) obtained by LA-ICP-MS.

As mentioned before under the discussion of the Bolt's
[5] Cave data, the $^{204}$Pb measurements produced by this technique are imprecise and potentially inaccurate. In this case, $^{208}$Pb was measured and can be used as a substitute for $^{204}$Pb. Th / U ratios (also measured by LA-ICP-MS) were extremely low, allowing us to ignore the radiogenic $^{208}$Pb
[10] contribution.

The uncorrected $^{206}$Pb / $^{238}$U isochron age of sample SB-1625-22 is $2.66 \pm 0.10$ Ma and exhibits significant overdispersion with respect to the analytical uncertainties (MSWD $= 36$). Model-3 isochron regression (sensu Ver-
[15] meesch, 2024) indicates that this excess scatter is equivalent to an age dispersion of $104 \pm 28$ kyr (Fig. 9a). In reality, the excess dispersion around the isochron is unlikely to reflect diachronous isotopic closure. A more likely explanation for the scatter of the $^{206}$Pb / $^{238}$U data is spatial variability of the
[20] $[4/8]_i$ values, as discussed in Sect. 5. In summary, the actual dispersion estimate probably has no physical meaning, but the isochron age should be as accurate as mathematically possible. The reduction of the scatter around the Botovskaya isochrons (Fig. 9a and c) towards the $y$ intercept also sug-

gests that the common-Pb ratio is not substantially correlated [25] with the postulated heterogeneous initial disequilibrium.

Given the antiquity of the sample and the difficulty of measuring $[4/8]_m$ by LA-ICP-MS, no initial disequilibrium measurement was made. Switching from $^{206}$U / $^{238}$U to $^{207}$Pb / $^{235}$U isochron space lowers the age to $1.60 \pm 0.10$ Ma [30] whilst reducing the dispersion of the data around the isochron line (MSWD $= 1.5$, Fig. 9b). To verify the accuracy of this result, it is useful to point out that a $[4/8]_i$ value of 3.9 would bring the corrected $^{206}$Pb / $^{238}$U isochron in alignment with the $^{207}$Pb / $^{235}$U isochron. Such a value is consistent with [35] the initial activity ratios of the more recent Botovskaya deposits (Fig. 7b). This not only supports the accuracy of the $^{207}$Pb / $^{235}$U isochron results but also suggests that the $[4/8]_i$ ratios have remained stable over hundreds of thousands of years. [40]

We would like to conclude this section by drawing attention to the fact that the $^{207}$Pb / $^{235}$U method is not always successful. Figures 9c and d show that Botovskaya sample SB-72-8 produces a well-defined linear array in $^{206}$Pb / $^{238}$U isochron space but fails to do so in $^{207}$Pb / $^{235}$U isochron [45] space. Such cases are not rare. The $^{207}$Pb / $^{235}$U approach only works in samples that are sufficiently rich in U and sufficiently poor in common Pb. Speleothems from Botovskaya

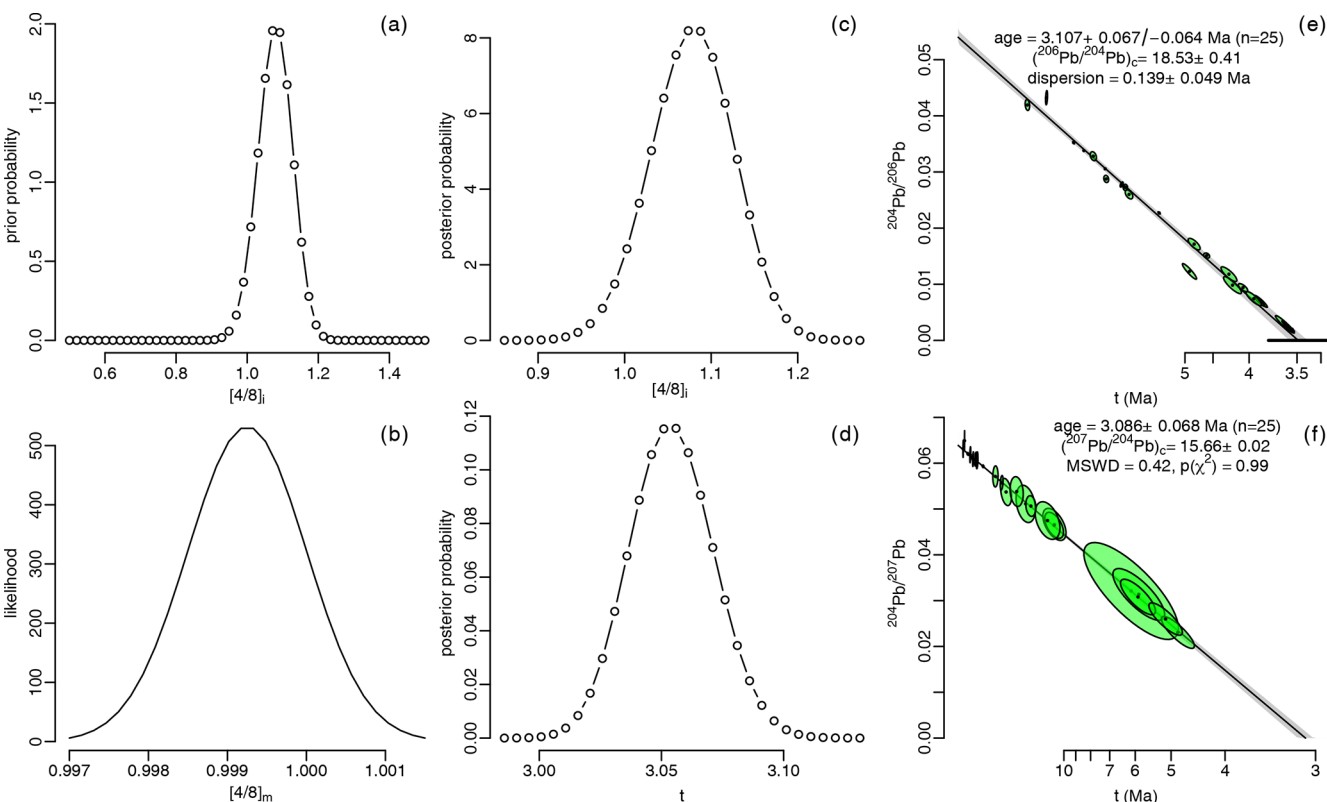

**Figure 8.** ID-TIMS U–Pb data for flowstone ASH-15K (Nuriel et al., 2021). **(a)** Informative prior for $[4/8]_i$, based on the data compilation of Chaldekas et al. (2022); **(b)** likelihood function for $[4/8]_m$, using the activity ratio measurements of ASH-15C1 (Vaks et al., 2013a); **(c, d)** posterior probabilities for $[4/8]_i$ and $t$, respectively; **(e)** the $^{206}Pb / ^{238}U$ isochron, fitted using the model-3 algorithm of Vermeesch (2024), with the excess dispersion shown as a horizontal 95 % error bar; and **(f)** the model-1 $^{207}Pb / ^{235}U$ isochron. The grey uncertainty bands represent the standard errors of the isochron fits and do not reflect the Bayesian credible intervals.

are rich in U (30–170 ppm for SB-72-8), so that here the problem seems to originate from common Pb (0.4–4 ppm for SB-72-8).

## 8  Implementation in `IsoplotR`

All the methods described in this paper have been implemented in the `IsoplotR` toolbox for geochronological data processing (Vermeesch, 2018). The matrix exponential disequilibrium correction method of Sect. 2 has been part of `IsoplotR` since version 3.0, whereas the deterministic Bayesian uncertainty estimation routine of Sect. 4 was introduced in version 5.2. At the time of writing, `IsoplotR` (version 6.7) supports 12 different U–Pb data formats. Disequilibrium corrected U–Pb isochron regression is available for all these formats, in different forms.

Formats 1–3 contain neither $^{204}Pb$ nor $^{208}Pb$. Therefore, isochron regression for these formats must be done by semitotal-Pb / U regression in Tera–Wasserburg concordia space. Formats 4–6 include $^{204}Pb$ as a common-Pb tracer. These formats permit the calculation of both $^{206}Pb / ^{238}U$ and $^{207}Pb / ^{235}U$ isochrons, either jointly (by

three-dimensional total-Pb / U isochron regression; Ludwig, 1998) or separately. To take full advantage of the $^{207}Pb / ^{235}U$ method's superior accuracy, it is recommended to use the two-dimensional option. Formats 7 and 8 use $^{208}Pb$ as a common-Pb tracer. They are also amenable to both $^{206}Pb / ^{238}U$ and $^{207}Pb / ^{235}U$ isochron regression, either jointly (by total-Pb / U–Th regression; Vermeesch, 2020) or separately. Formats 9–10 and 11–12 are simplified versions of formats 4–6 and 7–8, respectively, which only permit two-dimensional regression. Formats 9 and 11 are meant for $^{206}Pb / ^{238}U$ isochron regression, whereas formats 10 and 12 are meant for $^{207}Pb / ^{235}U$ isochron regression.

The disequilibrium corrections can be accessed from `IsoplotR`'s GUI (either online or offline) by using the "isochron" function and ticking the "apply disequilibrium correction" check box in the options menu. Alternatively, the same functionality can also be accessed from the command-line API. Bayesian uncertainty estimation is possible using either interface, but visualizing the posterior distributions of the parameter space is currently only possible from the command line.

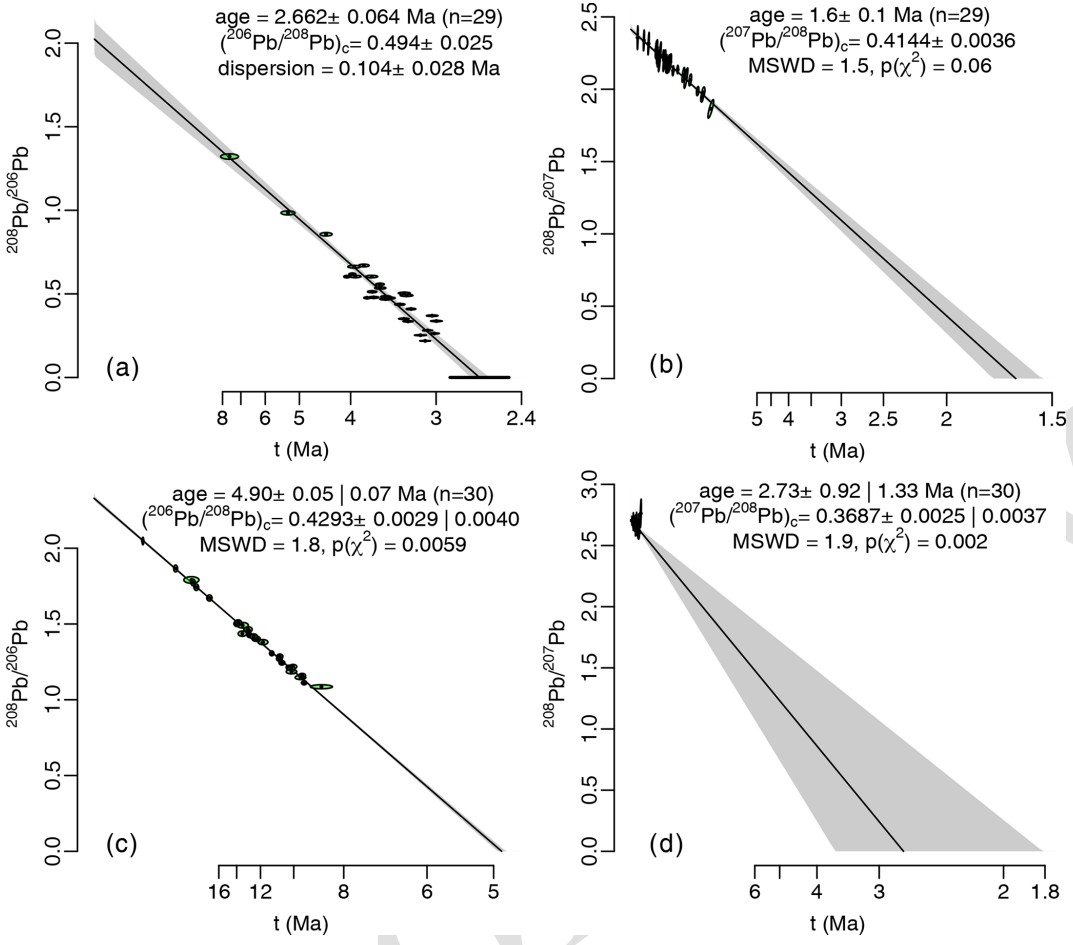

**Figure 9.** LA-ICP-MS data for two speleothems from Botovskaya cave in Siberia. **(a)** Model-3 $^{206}$Pb / $^{238}$U isochron regression for sample SB-1625-22. The equivalent model-1 isochron age (with MSWD = 9) is $2.65 \pm 0.10$ Ma. **(b)** Model-1 $^{207}$Pb / $^{235}$U isochron for SB-1625-22; **(c)** model-1 $^{206}$Pb / $^{238}$U, and **(d)** $^{207}$Pb / $^{235}$U isochrons for sample SB-72-8 ($2.73 \pm 1.33$ Ma).

The Supplement provides all the R code that was used to reproduce the figures in this paper (Vermeesch, 2025).

## 9 Conclusions

In this paper, we presented a critical appraisal of carbonate U–Pb geochronology and proposed three improvements to the technique. First, we introduced a matrix exponential solution to the initial disequilibrium problem, extending the work of Albarède (1995). This formulation produces identical results to the conventional solution by Engel et al. (2019) but can be written out more succinctly and can more easily be modified to suit other problems. For example, the matrix exponential approach can be adjusted to calculate disequilibrium-corrected U–Th–He ages (Farley et al., 2002; Danišík et al., 2017). Second, we presented a deterministic Bayesian algorithm to quantify the statistical uncertainty associated with the disequilibrium correction. This algorithm was used to demonstrate that, for samples older than ca. 2 Ma, disequilibrium-corrected $^{206}$Pb / $^{238}$U

geochronology is unreliable. Third, we advocated the use of the $^{207}$Pb / $^{235}$U isochron method as a more accurate alternative to the $^{206}$Pb / $^{238}$U method.

Although our findings are most relevant to young carbonates, the inaccuracy of the $^{206}$Pb / $^{238}$U method equally applies to old samples. Only the relative difference between the $^{206}$Pb / $^{238}$U and $^{207}$Pb / $^{235}$U ages reduces with time. The absolute difference remains constant at up to 4 Myr (Eq. 1). The corresponding systematic uncertainty cannot be removed without making unverifiable assumptions about the initial $^{234}$U / $^{238}$U activity ratio. For samples older than $> 100$ Ma, say, the systematic error caused by initial disequilibrium is generally smaller than the random errors associated with the isotope ratio measurements. However, given a sufficiently precise set of isochrons, it is theoretically possible to reconstruct the U disequilibrium conditions at the time of isotopic closure from the difference between the $^{206}$Pb / $^{238}$U and $^{207}$Pb / $^{235}$U clocks.

Engel et al. (2019) advocate using the same procedure in Quaternary studies. They propose a two-step pro-

cedure, whereby the difference between the $^{206}$Pb$\,/\,^{238}$U and $^{207}$Pb$\,/\,^{235}$U dates is used to estimate $[4/8]_i$; and this $[4/8]_i$ value is then used to calculate a corrected $^{206}$Pb$\,/\,^{238}$U age. IsoplotR implements a one-step algorithm that achieves the same goal using the total-Pb$\,/\,$U algorithm of Ludwig (1998) and the total-Pb$\,/\,$U–Th algorithm of Vermeesch (2020). However, we would like to add a note of caution about the usefulness of this joint regression procedure. Beyond ca. 2 Ma, all the age-resolving power of the paired $^{206}$Pb$\,/\,^{238}$U and $^{207}$Pb$\,/\,^{235}$U approach resides in the $^{207}$Pb$\,/\,^{235}$U clock, so the $^{206}$Pb$\,/\,^{238}$U data add no value.

**Code and data availability.** IsoplotR is free software released under the GPL-3 licence. The package and its source code are available from https://cran.r-project.org/package=IsoplotR (Vermeesch, 2018). The test data used in Sect. 8 are provided at https://doi.org/10.5281/zenodo.16262131 (Vermeesch, 2025).

**Author contributions.** NM developed the matrix exponential solution to the initial disequilibrium problem (McLean et al., 2016); PV developed the deterministic Bayesian inversion algorithm; RP proposed the $^{207}$Pb$\,/\,^{235}$U fix to the $^{206}$Pb$\,/\,^{238}$U problems; AV, TG, and SB provided the Siberian data; and PV wrote the paper, with feedback from the other authors.

**Competing interests.** At least one of the (co-)authors is a member of the editorial board of *Geochronology*. The peer-review process was guided by an independent editor, and the authors also have no other competing interests to declare. TS6

**Acknowledgements.** This paper benefited from thorough reviews by Perach Nuriel, Timothy Pollard, and Robyn Pickering.

**Financial support.** This research has been supported by the Natural Environment Research Council (grant no. NE/T001518/1, awarded to Pieter Vermeesch TS7) and the Leverhulme Trust (grant no. RPG-20202-334, awarded to Sebastian F. M. Breitenbach TS8). TS9

**Review statement.** This paper was edited by Axel Schmitt and reviewed by Perach Nuriel and Robyn Pickering.

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

## Remarks from the language copy-editor

## Remarks from the typesetter

**TS2**  Since certain fonts or other content are often not embedded in vector graphics, such content would not show up in some browsers or *.pdf viewers. To stay inclusive and allow all readers of our publications to make use of the article *.pdf files, we decided to include all figures as *.png or *.jpg files in PDFLaTeX instead of using the authors' *.eps, *.ps, or *.pdf vector graphics. However, since we also publish all articles in full-text HTML, we will provide your vector graphics as high-resolution figures so that readers are able to download and enlarge the figures for re-use (see e.g. https://www.hydrol-earth-syst-sci.net/23/1163/2019/). Please note that this high-resolution download is only possible if your figure has the Creative Commons Attribution 4.0 License (CC BY) applied. This is the case for the figures compiled by you or your co-authors. If you cite a figure from another paper that is not distributed under the Creative Commons Attribution License, the figure is identified as protected and the download link will be hidden.