# Peer review of "Carbonate 206Pb/238U problems and potential 207Pb/235U fixes"

_EGUsphere, 2025_

## Community Comment (CC1)

**Community comment on 'Broken $^{206}$Pb/$^{238}$U carbonate chronometers and $^{207}$Pb/$^{235}$U fixes' by Vermeesch et al.**

Timothy Pollard
School of Geography, Earth and Atmospheric Sciences, University of Melbourne
timothy.pollard@unimelb.edu.au

This manuscript presents a new methodology for calculating disequilibrium-corrected U–Pb ages and age uncertainties for young carbonate samples using a Bayesian statistics approach. It also offers a number of criticisms of existing $^{206}$Pb/$^{238}$U-based methods for dating young carbonates, going as far as to suggest that '…one could argue that the $^{206}$Pb/$^{238}$U method is of limited use to carbonate U–Pb geochronology'. Following this, the manuscript restates arguments presented in previous publications recommending use of $^{207}$Pb/$^{235}$U-based methods where accurate correction for initial [34/38] [1] disequilibrium via a measured [34/38] value is not possible.

While their Bayesian approach to calculating U–Pb ages for carbonate samples lying just outside the bounds of analytically resolvable disequilibrium is a welcome contribution, there are some inaccuracies in the arguments provided against use of $^{206}$Pb/$^{238}$U-based methods more generally. Furthermore, the authors fail to properly acknowledge and cite previous work undertaken on some key issues discussed in the manuscript, including work which advocates very similar ideas. This community comment addresses some of these issues.

**Monte Carlo approach**

Line 90 states incorrectly that `Isoplot` (Ludwig, 2003b) implements a Monte Carlo approach for estimating uncertainties in disequilibrium-corrected U–Pb ages that incorporate a measured [34/38] value, before going on to criticise this approach. This is both factually incorrect and somewhat unfair to Ludwig. `Isoplot` includes functions for calculating $^{206}$Pb*/$^{238}$U and $^{207}$Pb*/$^{235}$U ratios (where the * superscript denotes radiogenic Pb) as a function of age and initial intermediate 'daughter' activity ratios. These functions, in combination with the $^{234}$U-age equation, can be used to calculate disequilibrium-corrected U–Pb ages using a measured [34/38] value and estimate uncertainties via Monte Carlo simulation as part of a spreadsheet-based approach (Woodhead et al., 2006). However, this functionality is not part of `Isoplot`, nor is it advocated or discussed in the `Isoplot` manual, which is very clear in outlining the program's functionality.

Section 3 also provides a misleading description of the Monte Carlo approach implemented in `DQPB`. Prior to beginning the Monte Carlo simulation, `DQPB` checks that the measured [34/38] value provided by the user is statistically distinguishable from secular equilibrium. By default, the [34/38] value must be 2-σ removed from secular equilibrium, although this cut-off value can be adjusted. Where the measured [34/38] value fails this test, a warning is provided stating that the results of the Monte Carlo simulation may be inaccurate and should not proceed. The user is then encouraged to cancel the simulation. This is outlined in section 5.1 of Pollard et al. (2023).

Line 105 states that the quasi Monte Carlo approach of Vermeesch et al. is '…faster and produces deterministic results that do not depend on the seed of a random number generator.' In reality the difference in speed is of no practical significance on modern computers and the indeterminate nature of Monte Carlo simulation is trivial provided a sufficient number of trials are performed. There is also no reason to assume that deterministic uncertainty estimates are inherently better than non-deterministic ones: at the end of the day they are both just estimates, and neither is perfectly accurate. A more
* * *
[1] The notation of Vermeesch et al. is used throughout.

substantial problem, however, is that the quasi Monte Carlo approach appears to introduce spurious noise in some cases. For example, the arbitrary fluctuations in the probability density curves in Figure 2c and 5d are hardly realistic. This potential drawback should be discussed along with the apparent advantages of this approach.

**The matrix exponential approach**

There are numerous methods that can be used to solve the system of linear first-order differential equations describing the U–Pb radioactive decay series (or part thereof). These include:

- The original solution of Bateman (1910), which was applied directly to the U–Pb decay series by Ludwig (1977)[2].

- Methods that employ the Laplace transform explicitly (e.g. Catchen, 1984; Pressyanov, 2002; Bourdon et al., 2003).

- The matrix exponential approach, which has been used in nuclear science and engineering for over 50 years (e.g. Bell, 1973), and has been applied previously to the U–Pb decay series by Albarède (1995)[3].

These are all mathematically valid approaches, and, as such, should yield identical results for the same inputs (at least, for the simplified form of the U–Pb decay series considered in geochronology where numerical errors are a minor concern). However, this pre-existing literature is not discussed at all in the manuscript and the matrix exponential approach is presented as if it is an entirely novel idea. Given the well-established mathematical credentials of the authors (e.g. Mclean, 2014), it is certainly possible that they derived the matrix exponential equations independently and without knowledge of previous work on this topic. Nevertheless, it is typical practice in academic writing to cite previous work that presents the same or very similar ideas. Some clarification is required here.

This section also contains no substantial discussion of why the matrix exponential approach is chosen here or how it improves on other more widely used pre-existing solutions (e.g., Ludwig, 1977). Perhaps the argument is that the matrix exponential approach is more 'elegant' than other approaches. However, this is rather subjective and debatable given that the matrix exponential approach, at least as implemented in `IsoplotR`, requires more computations than simpler approaches to obtain the same result (at least for age calculations of the type discussed in the manuscript). There may be some advantages in terms of flexibility, however, if so, these should be outlined explicitly.

**Maximum range of measurable [$^{234}$U/$^{238}$U] disequilibrium**

Section 5 discusses the maximum range at which measured [34/38] is analytically resolvable from equilibrium. Two choices are made here that lead to an overly pessimistic assessment of this maximum range. The first is the adoption of an analytical uncertainty of 0.004 (2σ)[4], i.e. 4‰, for measured [34/38] values. Analytical uncertainties for [34/38] measurements that are made using modern MC-ICP-MS protocols with the $^{234}$U$^+$ beam collected using an ion counter typically range between 1–3 ‰ (2σ)  (e.g. Hellstrom, 2003; Shen et al., 2012; Cheng et al., 2013). However, it is possible to obtain analytical
* * *
[2] Note there is a typo in Equation (4) of Ludwig (1977). The $e^{\lambda_{234}t}$ term should read $e^{\lambda_{238}t}$.

[3] Albarède does not consider in-growth of stable $^{206}$Pb but the same mathematical approach can be adapted for calculations including radiogenic Pb.

[4] Strictly speaking, Vermeesch et al. adopt a relative analytical uncertainty of 0.002 (1σ). However, given measured values are always close to one in this context, this is essentially the same as adopting an absolute uncertainty of 0.004 (2σ).

uncertainties that are almost an order or magnitude lower than this (i.e. 0.2–0.3 ‰) using an all-Faraday cup analytical protocol (e.g. Andersen et al., 2004; Cheng et al., 2013, 2016; Kerber et al., 2023). Repeating the calculations in Table 1 using a [34/38] analytical uncertainty consistent with these high-precision all-Faraday protocols, which are clearly more appropriate for analysing samples approaching the limit of measurable disequilibrium, leads to significantly different results.

Perhaps the authors would justify use of a 4‰ (2σ) uncertainty based on arguments put forward in the Vermeesch et al. (2025) preprint—cited multiple times throughout the manuscript—which claims that speleothem [34/38] measurements are, as a general rule, overdispersed with respect to their analytical uncertainties. For example, this line of argument is suggested on line 230 where they write '…making the optimistic assumption that the analytical uncertainty of the $[34/38]_m$-measurements faithfully captures all sources of dispersion'. However, the arguments put forward by Vermeesch et al. (2025) on this matter are deficient in a number of ways:

1. Firstly, they erroneously state that the $[34/38]_m$ analyses from Walker et al. (2006) come from '*a* Sterkfontein speleothem' (my emphasis), and refer to them as 'duplicate samples', implying they are also representative of the same growth interval. However, it is very clear from the text in Walker et al. (2006), as well as from Figure 1, that the [34/38] measurements actually come from at least three[5] separate flowstones: STA09, STA12, and STA15. These flowstones were not even all collected from the same horizon, and so it is in no way surprising that their combined $[34/38]_m$ data are overdispersed with respect to their analytical uncertainties. It is also not clear how the sampling was performed for each individual flowstone, i.e. were the sub-samples carefully extracted from the same growth layer, or were they sampled across different growth zones? This detail is critical in assessing overdispersion because it is very well known that speleothem $[34/38]_i$ values can vary considerably across growth layers due to factors such as changing hydrological conditions (e.g. Fairchild & Treble, 2009). Therefore, great care needs to be taken in sampling young speleothems for U/Pb and [34/38] analysis to ensure each aliquot is representative of the same growth interval.

2. Beyond their misinterpretation of the $[34/38]_m$ data from Walker et al. (2006), Vermeesch et al. (2025) present a logically flawed argument. They assert that because, according to their assessment, the $[34/38]_m$ data from Walker et al. (2006) are overdispersed relative to analytical uncertainties, then all speleothem $[34/38]_m$ ratios are likely to be similarly overdispersed, rendering U–Pb ages and uncertainties derived from measured $[34/38]_m$ values inherently unreliable. However, this reasoning relies on an extreme extrapolation from a single dataset to the global population of speleothems. This is analogous to claiming that because one isochron dataset of type x is overdispersed, all isochrons of type x must be as well. While some speleothems may indeed show heterogeneous [34/38] ratios along individual growth increments (e.g., due to episodic U-loss), this is far from a universal phenomenon as evidenced by the large body of speleothem U–Th datasets, which show that, for many speleothems, age inversions due to significant U-loss or gain are relatively rare.

3. Finally, Vermeesch et al. (2025) present some new speleothem $[34/38]_m$ data from a South African flowstone in Figure 6ii, which appear to be overdispersed with respect to their analytical uncertainties. However, these data are incomplete and cannot be properly assessed because no details of the sampling or analytical method are provided by the authors.

For these reasons, reference to the Vermeesch et al. (2025) pre-print should not be accepted as justification for use of a 4 ‰ (2σ) $[34/38]_m$ uncertainty here. It may be helpful if the authors consider
* * *
[5] It is not clear how STA14 fits into the picture, but it is probably a fourth flowstone.

two different scenarios in this section: one assuming that a typical ion counter protocol for [34/38] measurements is adopted and the other assuming a high-precision all-Faraday protocol is used.

The second choice made in section 5 that leads to a particularly pessimistic assessment of the maximum range of measurable $[34/38]_m$ disequilibrium is the adoption of a 3-$\sigma$ cut-off to define what is 'statistically distinguishable'. This is at odds with common practice in geochronology, where statistical tests and age calculation results are near-ubiquitously reported at the 2-$\sigma$ level (or equivalently using a probability threshold of 0.05). For example, it seems that essentially all results provided by `Isoplot` (Ludwig, 2012) and `IsoplotR` (Vermeesch, 2018) are presented in this way. There may be some justification for using a 3-$\sigma$ cut-off here and not in other geochronological calculations, but if so, this should be provided.

**Bayesian approach**

The Bayesian methodology introduced in Section 4 seems to be a reasonable approach to handling the situation where a suitable $[34/348]_m$ value is available, but it marginally overlaps secular equilibrium, making standard age uncertainty propagation approaches unreliable. However, the default $[34/48]_i$ prior that is proposed, i.e. a uniform distribution spanning 0 to 20, is not at all consistent with the available global data for speleothems (e.g. Hellstrom, 2013; Markowska et al., 2025). For one thing, there is far too much $[34/48]_i$ probability density assigned to the interval 10–20. Therefore, it is not clear that use of this prior would produce accurate results.

In most settings a more informative local or regional prior could be constructed. It is stated on line 132 that such a prior could easily be accommodated by the Bayesian algorithm, but there is no discussion of how this might influence the calculated age or its age uncertainties relative to use of the uniform prior, which is proposed as the default.

Finally, this age calculation approach yields age uncertainty estimates in the form of Bayesian credible intervals instead of frequentist-type confidence intervals, which are more commonly encountered in geochronology. Perhaps it would be useful to include a discussion of how these Bayesian credible intervals should be interpreted, and what the practical differences are between credible intervals and confidence intervals in this geochronological context?

**Limits of U–Th dating method**

Line 20 states that the limit of the U–Th disequilibrium dating method is 800 ka, whereas line 204 implies it is ~1 Ma. These limits are obviously inconsistent and there are no arguments or citations provided to support either.

U–Th age uncertainties tend to increase in an approximately exponential manner with age, and become significantly degraded and skewed as the system approaches secular equilibrium (e.g. Ludwig, 2003a). The nominal limit of the U–Th method depends to some extent on the amount of age uncertainty that can be accommodated in usefully addressing a particular scientific issue. However, there is also a hard limit at the point where the measured [34/38]-[30/38] composition of a sample becomes indistinguishable from infinite age compositions, i.e. where the measured [34/38]-[30/38] composition significantly overlaps the 'infinite' age isochron within uncertainty on a [34/38] versus [30/38] evolution diagram (Woodhead et al., 2019). At this point, the method is only useful for providing a reliable lower age bound.

Using the authors' preferred [34/38] measurement uncertainty of ±4 ‰ (2$\sigma$), and assuming a similar analytical precision for $[30/38]_m$, along with an initial [34/38] value of 1.2, U–Th age uncertainties are approximately –69/+174 at 550 ka. An age with this level of relative uncertainty is

unlikely to be useful for addressing any key scientific issues in Quaternary science. Not far beyond 550 ka, the upper age bound becomes undefined for this level of analytical precision.

If activity ratio measurement uncertainties more consistent with high-precision U–Th measurement protocols are adopted, then the useful range of the U–Th method may is extended beyond 600 ka (e.g. Cheng et al., 2016). However, there is certainly no argument that this limit is ~1 Ma.

**[208]Pb normalization**

Section 6 discusses the normalisation of Pb and U isotopic ratios to [208]Pb instead of [204]Pb when dating young carbonates. This idea has already been advocated in several previous studies that should be properly cited (e.g. Getty et al., 2001; Walker et al., 2006; Parrish et al., 2018; Engel et al., 2019).

**ASH-15 isochron fit**

Section 7.2 provides a re-assessment of data previously published for the ASH-15 flowstone, which has been proposed as a suitable reference material for carbonate U–Pb dating by LA-ICPMS (Nuriel et al., 2021). Part of this re-assessment involves fitting an isochron line to the TIMS $^{206}$Pb/$^{204}$Pb-$^{238}$U/$^{204}$Pb data of Nuriel et al. using the `model-3` algorithm of Vermeesch (2024). It is not clear whether the '3a' or '3b' version of this algorithm is employed here because this detail is omitted.

The `model-3a` algorithm attributes all excess dispersion (i.e. dispersion above that which is accounted for by analytical uncertainties) to variability in the inherited Pb component, whereas the `model-3b` algorithm attributes all excess dispersion to 'diachronous isotopic closure' of the sub-samples. In both cases the excess dispersion is attributed to a particular cause and is assumed to follow a strictly Gaussian distribution. These are both strong assumptions, in the sense that they can exert a significant impact on the resulting isochron best-fit line and its uncertainty. Therefore, when implementing these algorithms, there is a clear need to rigorously justify these assumptions.

In this case, it is difficult to envisage how the assumption of 'diachronous isotopic closure' could apply, unless the flowstone grew unusually slowly, and the samples were inappropriately extracted from different growth zones. On the other hand, the assumption of a heterogenous initial Pb isotopic composition, which was suggested as the primary cause of overdispersion in the original publication, seems more reasonable. However, even if this is the only cause of overdispersion, it is not clear why variability in the inherited-Pb isotopic ratio would necessarily follow a strictly Gaussian distribution; it seems as if it could conceivably follow a range of different statistical distributions.

In addition to a heterogenous isotopic composition of inherited Pb, there are other possible causes of overdispersion that should be considered. These include:

1. Pb contamination: either varying amounts of Pb contamination or contamination with Pb components that have different isotopic compositions. This possibility should be considered here because the sub-samples analysed by Nuriel et al. (2021) were not subjected to leaching in dilute HCl prior to chemistry, which previous studies have shown is crucial to removing Pb contamination introduced during sample handling and preparation, which can easily dominate the overall Pb budget in young carbonates (e.g. Woodhead et al., 2006, 2012).

2. Open-system behavior, and especially U-loss. This may impart a similar effect to 'diachronous isotopic closure' but is unlikely to conform to a strict Gaussian distribution.

Whatever assumptions are employed here, it is essential that they are stated clearly and rigorously justified.

Finally, contrary to the what is written in the *Code and data availability* declaration, the 'ASH15K' data are not provided in the supplementary information.

**References**

Albarède, F. (1995). *Introduction to Geochemical Modeling*. Cambridge University Press. https://doi.org/10.1017/CBO9780511622960

Andersen, M. B., Stirling, C. H., Potter, E.-K., & Halliday, A. N. (2004). Toward epsilon levels of measurement precision on $^{234}U/^{238}U$ by using MC-ICPMS. *International Journal of Mass Spectrometry*, *237*(2), 107–118. https://doi.org/10.1016/j.ijms.2004.07.004

Bateman, H. (1910). Solution of a system of differential equations occurring in the theory of radioactive transformations: Proceedings of the Cambridge Philosophical Society. *Proceedings of the Cambridge Philosophical Society*, *15*, 423–427.

Bell, M. (1973). *ORIGEN: the ORNL isotope generation and depletion code* (No. ORNL-4628). Oak Ridge National Laboratory (ORNL), TN, USA. https://inis.iaea.org/records/mpm97-77602

Bourdon, B., Henderson, G. M., Lundstrom, C. C., & Turner, S. P. (2003). Introduction to U-series geochemistry. In B. Bourdon, S. Turner, G. M. Henderson, & C. C. Lundstrom (Eds.), *Uranium-series geochemistry* (Vol. 52, pp. 1–25). Mineralogical Society of America. https://pubs.geoscienceworld.org/msa/rimg/article-abstract/52/1/631/87473

Catchen, G. L. (1984). Application of the equations of radioactive growth and decay to geochronological models and explicit solution of the equations by Laplace transformation. *Chemical Geology*, *46*(3), 181–195. https://doi.org/10.1016/0009-2541(84)90188-8

Cheng, H., Edwards, R. L., Sinha, A., Spötl, C., Yi, L., Chen, S., Kelly, M., Kathayat, G., Wang, X., Li, X., Kong, X., Wang, Y., Ning, Y., & Zhang, H. (2016). The Asian monsoon over the past 640,000 years and ice age terminations. *Nature*, *534*(7609), 640–646. https://doi.org/10.1038/nature18591

Cheng, H., Lawrence Edwards, R., Shen, C.-C., Polyak, V. J., Asmerom, Y., Woodhead, J., Hellstrom, J., Wang, Y., Kong, X., Spötl, C., Wang, X., & Calvin Alexander, E. (2013). Improvements in $^{230}Th$ dating, $^{230}Th$ and $^{234}U$ half-life values, and U--Th isotopic measurements by multi-collector inductively coupled plasma mass spectrometry. *Earth and Planetary Science Letters*, *371–372*, 82–91. https://doi.org/10.1016/j.epsl.2013.04.006

Engel, J., Woodhead, J., Hellstrom, J., Maas, R., Drysdale, R., & Ford, D. (2019). Corrections for initial isotopic disequilibrium in the speleothem U-Pb dating method. *Quaternary Geochronology*, *54*, 101009. https://doi.org/10.1016/j.quageo.2019.101009

Fairchild, I. J., & Treble, P. C. (2009). Trace elements in speleothems as recorders of environmental change. *Quaternary Science Reviews*, *28*(5–6), 449–468. https://doi.org/10.1016/j.quascirev.2008.11.007

Getty, S. R., Asmerom, Y., Quinn, T. M., & Budd, A. F. (2001). Accelerated Pleistocene coral extinctions in the Caribbean Basin shown by uranium-lead (U-Pb) dating. *Geology*, *29*(7), 639–642. https://doi.org/10.1130/0091-7613(2001)029<0639:APCEIT>2.0.CO;2

Hellstrom, J. (2003). Rapid and accurate U/Th dating using parallel ion-counting multi-collector ICP-MS. *Journal of Analytical Atomic Spectrometry*, *18*(11), 1346. https://doi.org/10.1039/b308781f

Hellstrom, J. (2013). $^{234}U/^{238}U$ in speleothems revisited: Are there generally applicable relationships of this proxy to past environmental change? *Mineralogical Magazine*, *77*, 1282. https://doi.org/0.1180/minmag.2013.077.5.8

Kerber, I. K., Arps, J., Eichstädter, R., Kontor, F., Dornick, C., Schröder-Ritzrau, A., Babu, A., Warken, S., & Frank, N. (2023). Simultaneous U and Th isotope measurements for U-series dating using MCICPMS. *Nuclear Instruments and Methods in Physics Research Section B: Beam Interactions with Materials and Atoms*, *539*, 169–178. https://doi.org/10.1016/j.nimb.2023.04.003

Ludwig, K. R. (1977). Effect of initial radioactive-daughter disequilibrium on U-Pb isotope apparent ages of young minerals. *Journal of Research of the US Geological Survey*, *5*(6), 663–667.

Ludwig, K. R. (2003a). Mathematical–Statistical treatment of data and errors for $^{230}$Th/U geochronology. In B. Bourdon, S. Turner, G. M. Henderson, & C. C. Lundstrom (Eds.), *Uranium-series geochemistry* (Vol. 52, pp. 631–656). Mineralogical Society of America. https://pubs.geoscienceworld.org/msa/rimg/article-abstract/52/1/631/87473

Ludwig, K. R. (2003b). *User's manual for Isoplot 3.00: A geochronological toolkit for Microsoft Excel* (No. 4; Berkeley Geochronological Center Special Publication).

Ludwig, K. R. (2012). *Isoplot/Ex Version 3.75: A Geochronological Toolkit for Microsoft Excel*. Berkeley Geochronology Center.

Markowska, M., Martin, A. N., Fohlmeister, J., Warken, S., Kaushal, N., Columbu, A., Novello, V. F., & Azevedo, V. (2025). *A global compilation of initial $^{234}U/^{238}U$ variability in speleothems using the SISAL database*. Climate Change: The Karst Record X conference, Cape Town (South Africa).

Nuriel, P., Wotzlaw, J.-F., Ovtcharova, M., Vaks, A., Stremtan, C., Šala, M., Roberts, N. M. W., & Kylander-Clark, A. R. C. (2021). The use of ASH-15 flowstone as a matrix-matched reference material for laser-ablation U − Pb geochronology of calcite. *Geochronology*, *3*(1), 35–47. https://doi.org/10.5194/gchron-3-35-2021

Parrish, R. R., Parrish, C. M., & Lasalle, S. (2018). Vein calcite dating reveals Pyrenean orogen as cause of Paleogene deformation in southern England. *Journal of the Geological Society*, *175*(3), 425–442. https://doi.org/10.1144/jgs2017-107

Pollard, T., Woodhead, J., Hellstrom, J., Engel, J., Powell, R., & Drysdale, R. (2023). DQPB: software for calculating disequilibrium U–Pb ages. *Geochronology*, *5*(1), 181–196. https://doi.org/10.5194/gchron-5-181-2023

Pressyanov, D. S. (2002). Short solution of the radioactive decay chain equations. *American Journal of Physics*, *70*(4), 444–445. https://doi.org/10.1119/1.1427084

Shen, C.-C., Wu, C.-C., Cheng, H., Edwards, R. L., Hsieh, Y.-T., Gallet, S., Chang, C.-C., Li, T.-Y., Lam, D. D., Kano, A., Hori, M., & Spötl, C. (2012). High-precision and high-resolution carbonate $^{230}$Th dating by MC-ICP-MS with SEM protocols. *Geochimica et Cosmochimica Acta*, *99*(C), 71–86. https://doi.org/10.1016/j.gca.2012.09.018

Vermeesch, P. (2018). IsoplotR: A free and open toolbox for geochronology. *Geoscience Frontiers*, *9*(5), 1479–1493. https://doi.org/10.1016/j.gsf.2018.04.001

Vermeesch, P. (2024). Errorchrons and anchored isochrons in IsoplotR. *Geochronology*, *6*(3), 397–407. https://doi.org/10.5194/gchron-6-397-2024

Vermeesch, P., Hopley, P., Roberts, N., & Parrish, R. (2025). *Geochronology of Taung and other southern African australopiths* [pre-print] (B. A. Wood, F. E. Grine, & H. B. Smith, Eds.). https://tinyurl.com/Taung2025

Walker, J., Cliff, R. A., & Latham, A. G. (2006). U-Pb isotopic age of the StW 573 hominid from Sterkfontein, South Africa. *Science*, *314*(5805), 1592–1594. https://doi.org/10.1126/science.1132916

Woodhead, J. D., Sniderman, J. M. K., Hellstrom, J., Drysdale, R. N., Maas, R., White, N., White, S., & Devine, P. (2019). The antiquity of Nullarbor speleothems and implications for karst palaeoclimate archives. *Scientific Reports*, 1–8. https://doi.org/10.1038/s41598-018-37097-2

Woodhead, J., Hellstrom, J., Maas, R., Drysdale, R., Zanchetta, G., Devine, P., & Taylor, E. (2006). U–Pb geochronology of speleothems by MC-ICPMS. *Quaternary Geochronology*, *1*(3), 208–221. https://doi.org/10.1016/j.quageo.2006.08.002

Woodhead, J., Hellstrom, J., Pickering, R., Drysdale, R., Paul, B., & Bajo, P. (2012). U and Pb variability in older speleothems and strategies for their chronology. *Quaternary Geochronology*, *14*(C), 105–113. https://doi.org/10.1016/j.quageo.2012.02.028

---

## Author Comment (AC3)

**Further responses to the review by Robyn Pickering on "Broken $^{206}$Pb/$^{238}$U carbonate chronometers and $^{207}$Pb/$^{235}$U fixes"**

Pieter Vermeesch, Noah McLean,
Anton Vaks, Tzahi Golan,
Sebastian Breitenbach and Randall Parrish

This supplementary document (1) provides additions to our response to Dr. Pickering's review; (2) addresses comments that were left out of the main response for the sake of brevity; and (3) responds to personal attacks that distract attention from the scientific discussion.

> *To suggest that the entire 238-206 carbonate chronometer is 'broken' is a very bold claim and asks the geochronology community to throw out the last almost 20 years of work in this field, starting from Woodhead et al., 2006 but arguably 28 years of work if you start with Richards et al., 1998. This means disregarding a huge body of well-respected work on dating carbonates, mainly in this case speleothems, from around the world. Such a claim requires remarkable evidence, which this manuscript fails to deliver.*

We have already addressed this comment in the main response, but would also like to point out that the pioneering study of Richards et al. (1998) included $^{207}$Pb/$^{235}$U isochrons. Dr. Richards showed remarkable foresight to anticipate the problems reported in our paper.

As for the perceived lack of "remarkable evidence": no evidence is stronger than mathematical proof. Our paper provides a brand new mathematical tool to quantify the limitations of disequilibrium-corrected $^{206}$Pb/$^{238}$U-dating. This tool allows each and every geochronologist to assess whether their data fall in the 'danger zone' of unresolvable disequilibrium.

> *Further to this, Adams et al (2010) present palaeomagnetic data and a biochronological faunal analysis of Hoogland cave, which place the basal flowstone at ∼3.12 Ma, so the later Pickering et al (2019) direct dating of this same flowstone to 3.1 Ma is in complete concordance with the existing and independent geochronological data. So, basing the rubbishing of the entire 238-206 carbonate chronometer off this one sample is not only unwarranted but incorrect.*

In our opinion, reproducing a previously known age constraint does not provide a truly independent validation of the $^{206}$Pb/$^{238}$U-method. In any case, we do not know the age of the Hoogland sample, so we never claim that it is *not* 3.1 Ma. It is possible that the 3.1 Ma date is correct, by chance. $^{207}$Pb/$^{235}$U data would be helpful, but are lacking.

> *For the ID Siberian data, the authors here recalculate the 238-206 ages, using the published data for over 70 speleothem samples, and present matched 235-207 ages (their Figure 7). The 235-207 ages certainly do overlap with the corrected 238-206 ages but the errors on the 235-207 ages are huge by comparison, which is a serious detractor from this method. So yes, in this case, speleothems from this karst region have 238 and 235 ages which are coeval, but this is not a surprising nor novel results, we would expect this. The much larger errors on the 235 ages are a strike against this approach vs making an argument to use it in favour of the 238 ages.*

The choice between the $^{206}$Pb/$^{238}$U and $^{207}$Pb/$^{235}$U methods is a choice between precision and accuracy. The accuracy of the $^{206}$Pb/$^{238}$U method decreases with age, whereas the precision of the $^{207}$Pb/$^{235}$U method improves with age. The crossover point occurs between 1.5 and 3 Ma, depending on the sample. This is not controversial. It would be wrong to focus only on precision whilst neglecting accuracy.

> *The LA Siberian data is also looked at, as an example where measuring residual 234-238 was not done, as this is not really possible with LA (ID is the only way to get solid 234-238 measurements), so they argue that their 235 age is better, as it does not need this 234-238 'correction', which was not possible given that this is a LA dataset.*

The purpose of the LA example is not to compare the $^{206}$Pb/$^{238}$U method with the $^{207}$Pb/$^{235}$U method, but simply to demonstrate a potential application of the $^{207}$Pb/$^{235}$U method. As the reviewer points out, $^{206}$Pb/$^{238}$U dating requires $^{234}$U/$^{238}$U activity ratio measurements, using TIMS or HR-ICPMS. This expensive and time consuming step is not necessary for the $^{207}$Pb/$^{235}$U method. This is a major advantage of the $^{207}$Pb/$^{235}$U method over the $^{206}$Pb/$^{238}$U method. We will highlight this in the revised manuscript.

> *To be clear, the 234-238 measurements are not used by the U-Pb carbonate community as a 'correction' but part of the age calculations. The word 'correction' implies that this is used as an afterthought, and that the 238-206 ages need 'correcting'. The 234-238 measurements are a routine part of U-Pb age determinations, and are part of how the final age is calculated.*

This is a semantic discussion. We think that it is useful to compare the $^{206}$Pb/$^{238}$U dates with and without initial disequilibrium. We do not know how

else to call the difference between these two values than a 'correction'. When the difference is large, then this means that most of the time-resolving power lies in the $^{234}$U/$^{238}$U data and not in the $^{206}$Pb/$^{238}$U measurements. Thus, we would argue that it referring to the $^{234}$U/$^{238}$U-calculations as a 'correction' is helpful and protects users against pushing their data too far.

> *The same is true doing U-Th dating. So this is another almost meaningless case study, and to me shows the lack of familiarity these authors have with the routine work of U-series dating, rather than an intrinsic issue with the method.*

Personal attacks like this are not productive. The authors have decades worth of practical experience in geochronology, including U-series methods. The use of the word 'routine' may be a Freudian slip of the tongue. The authors have a strong track record in methodological innovation. The reviewer lacks this experience, which may explain her reluctance to question basic assumptions.

> *The new standard, ASH-15, does not record any initial 234-238 disequilibrium, so unsurprisingly, the 238-206 and 235-207 results are in near perfect agreement. This is not a useful case study here and not does illustrate in any way how the 238-206 chronometer is 'broken'.*

The purpose of this example was precisely to show how the $^{206}$Pb/$^{238}$U- and $^{207}$Pb/$^{235}$U-methods agree in areas without substantial initial disequilibrium. It appears that the use of the word 'broken' in the title of our paper has completely skewed the reviewer's perception of our work. We hope that the change of title will fix this problem.

We would like to note that Reviewer 1 (Perach Nuriel, original author of the ASH-15 dataset) approved of the second case study. Dr. Nuriel recognised that the $^{207}$Pb/$^{235}$U isochron is more accurate, whilst raising the important points that (1) the poorer accuracy of the $^{206}$Pb/$^{238}$U isochron does not degrade the value of ASH-15 as a reference material; and (2) disequilibrium effects do not only limit the accuracy of $^{206}$Pb/$^{238}$U-dates in the Plio-Pleistocene, but are equally relevant for older carbonates.

In conclusion, the second case study is useful and we want to keep it.

> *So, none of these three case studies present compelling evidence to abandon the 238 chronometer in favour of the 235 one. Further to this point, in this case study they go on to that the 235 chronometer only works in speleothems with very high U concentrations, the Siberian speleothems are reported to have between 30 and 170ppm of U, which is very high. Engel and Pickering (2022) look at the concentration of U in a much bigger dataset of geographically spread U-Pb ages (South Africa, Australia and Italy) and find much much lower U concentrations are the norm. Again, this speaks to these authors lack of familiarity with the norms and strategies of the U-Pb carbonate community.*

The applicability of the $^{207}$Pb/$^{235}$U method does not only depend on the uranium concentration. It also depends on the concentration of common lead. Roberts et al. (2020, with co-author McLean) show how LA-ICP-MS mapping can be used as a screening tool to find suitable samples. Vermeesch et al. (2025, with co-author Parrish) used this strategy to date a South African flowstone. In conclusion, the authors are not just familiar with "the norms and strategies of the U-Pb carbonate community", but actually help shape those norms and strategies.

> *Saying that it is safe to assume there is no detrital Th in 'clean' samples is another example – it is standard practise to apply a 232-230 correction to all U-Th age data and to use these as the final ages, regardless on the appearance of the sample.*

We are not sure exactly which assumption the reviewer is referring to here. There are two cases where we assume that the carbonate material contains no detrital Th:

1. When applying the $^{230}$Th 'correction' to $^{206}$Pb/$^{238}$U isochrons. Note that this is another weakness of the $^{206}$Pb/$^{238}$U method, which strengthens our case for the $^{207}$Pb/$^{235}$U method rather than weaken it. Note that the assumption of zero initial $^{231}$Pa has a much smaller effect on the final age estimate for the $^{207}$Pb/$^{235}$U method.

2. During the construction of $^{207}$Pb/$^{208}$Pb – $^{235}$U/$^{208}$Pb isochrons, we assume that the $^{208}$Pb is dominated by common Pb, and that radiogenic $^{208}$Pb can be neglected. This assumption is generally safe due to the long half life of $^{232}$Th (14 Ga). However, as mentioned in the response to Reviewer 1, it is possible to correct for the presence of Th, using the methods of Vermeesch (2020).

> *5. The first line in section 6 is also misleading (line 176): "In the previous section, we showed that the accuracy of the 206Pb/238U method is undermined by the extreme 234U-enrichment that is observed in some ground waters..." this is a massive overgeneralization, as we do not in fact see such enrichment that often. The Hoogland example is a case but this is literally just one case in what is now a huge, global U-Pb carbonate dataset and is not typical to what we normally see. So where they "show" that the accuracy is undermined is based on data that is not necessarily representative of a typical speleothem. This is yet another example of how unfamiliar these authors are with the standards and norms of this field.*

The paper reports not one but two areas where strong $^{234}$U-enrichment has taken place: South Africa ([4/8] < 12; Kronfeld et al., 1994) and Siberia ([4/8]$_i$ < 5; Vaks et al., 2020). A quick literature survey reveals that there are other places in the world where this is the case, such as Finland ([4/8]$_i$ < 4;

Asikainen, 1981) and Japan ($[4/8]_i < 11$; Kuribayashi et al., 2025). The cautious approach would be to assume a worst case scenario and work one's way back from there.

> *it is worth also noting, that none of the authors are themselves U-Pb carbonate specialists; there are certainly expertise in Ar-Ar data, geochronological data handling, U-Pb zirchon [sic] data, stalagmites and some lab technical skills but to make such a big claim about the 238 chronometer to have some validity, one would need to see it coming from experts in the field.*

These are personal attacks, not objective scientific arguments. They are wrong, too.

> *This is a small observation but last authors surname is misspelled (Parris vs Parish [sic]) which is not a good look.*

The manuscript uses the correct spelling of Randy Parrish's name. However, it is true that a typo was made in the online proforma document. Humans make mistakes. This is not a problem, as longs as we fix them. The same comment applies to carbonate U–Pb geochronology.

> *These authors go onto the argue that there they provide a better, faster new way of calculating U-Pb carbonate ages. Again this is not a strong argument and is based on their perception of an issue, which is not shared by the U-Pb carbonate community. The speed of the current age calculators is not an issue, no one is looking for a 'faster' method. Their only other argument for their method is that the existing routine by Engel (2019) 'can be written out more succinctly', which is an observation but not the basis on which to argue that we need a new method.*

The reviewer confuses two things here. Our re-implementation of Engel et al. (2019)'s algorithm is faster than the original Monte Carlo method. Our Bayesian inversion method is not particularly fast. In fact, it's probably slightly slower than the Monte Carlo method, especially when $^{230}Th/^{238}U$ data are included. As for our preference for matrix exponentials, whilst it is true that its succinctness is a personal preference, it also offers advantages such as the ease to incorporate additional intermediate daughters if so desired. The matrix exponential method can also be used for disequilibrium corrections in U–Th–He thermochronology. Thus, there is plenty of value in publishing the matrix exponential approach in *GChron*.

> *In summary, none of the arguments presented here are compelling and I can see no value in this manuscript being published, there is potential to do more harm than good here.*

We are unsure which 'harm' the reviewer is referring to. Our paper urges the reader to exercise caution when applying disequilibrium corrections to $^{206}$Pb/$^{238}$U data. Caution is a good thing.

Although the reviewer left no stone unturned (which, as mentioned before, is why we suggested her name to the editor), none of her arguments undermine our research findings. Despite the personal attacks, we have found the review useful because (1) surviving this barage of criticism strengthens our hypothesis in a Popperian sense; and (2) the review has highlighted a few points that would benefit from further clarification.

> *Previous work should be better cited, particularly for the use of 207Pb/235U and normalization with 208Pb*

Agreed, as discussed in our response to Reviewer 1 and the Community Comment.

> *Figure 5 caption: typo "206Pb/206U"*

Thanks. This will be fixed.

**References**

Asikainen, M. State of disequilibrium between $^{238}$U, $^{234}$U, $^{226}$Ra and $^{222}$Rn in groundwater from bedrock. *Geochimica et Cosmochimica Acta*, 45(2):201–206, 1981.

Engel, J., Woodhead, J., Hellstrom, J., Maas, R., Drysdale, R., and Ford, D. Corrections for initial isotopic disequilibrium in the speleothem U-Pb dating method. *Quaternary Geochronology*, 54:101009, 2019.

Kronfeld, J., Vogel, J., and Talma, A. A new explanation for extreme $^{234}$U/$^{238}$U disequilibria in a dolomitic aquifer. *Earth and Planetary Science Letters*, 123 (1-3):81–93, 1994.

Kuribayashi, C., Miyakawa, K., Ito, A., and Tanimizu, M. Large disequilibrium of $^{234}$U/$^{238}$U isotope ratios in deep groundwater and its potential application as a groundwater mixing indicator. *Geochemical Journal*, 59(2):35–44, 2025.

Richards, D. A., Bottrell, S. H., Cliff, R. A., Ströhle, K., and Rowe, P. J. U-Pb dating of a speleothem of Quaternary age. *Geochimica et Cosmochimica Acta*, 62(23-24):3683–3688, 1998.

Roberts, N. M., Drost, K., Horstwood, M. S., Condon, D. J., Chew, D., Drake, H., Milodowski, A. E., McLean, N. M., Smye, A. J., Walker, R. J., et al. Laser ablation inductively coupled plasma mass spectrometry (LA-ICP-MS) U–Pb carbonate geochronology: strategies, progress, and limitations. *Geochronology*, 2(1):33–61, 2020.

Vaks, A., Mason, A., Breitenbach, S., Kononov, A., Osinzev, A., Rosensaft, M., Borshevsky, A., Gutareva, O., and Henderson, G. Palaeoclimate evidence of vulnerable permafrost during times of low sea ice. *Nature*, 577(7789):221–225, 2020.

Vermeesch, P. Unifying the U–Pb and Th–Pb methods: joint isochron regression and common Pb correction. *Geochronology*, 2(1):119–131, 2020.

Vermeesch, P., Hopley, P., Roberts, N., and Parrish, R. Geochronology of Taung and other southern African australopiths. In Wood, B. A., Grine, F. E., and Smith, H. B., editors, *One hundred years of* Australopithecus africanus, chapter 4. Springer, 2025.

---

## Author Response (AR1)

Prof. Pieter Vermeesch
University College London
+44 (0)20 3108 6369
https://ucl.ac.uk/~ucfbpve/

2 July 2025

Dear Prof. Schmitt,

I hereby submit the revised manuscript on carbonate U–Pb geochronology, which addresses all the points raised in the public discussion. In this cover letter, I would like to highlight the most important changes, and address the additional points raised in your decision letter:

1. The title has been changed to "Carbonate $^{206}$Pb/$^{238}$U problems and potential $^{207}$Pb/$^{235}$U fixes".

2. We followed your suggestion (and that of Dr. Pollard) to give greater prominence to the informative priors. The revised manuscript applies a $^{234}$U/$^{238}$U disequilibrium correction to ASH-15, using a prior distribution obtained from an initial $^{234}$U/$^{238}$U activity ratio compilation of Chaldekas et al. (2022).

3. The Hoogland example has been replaced with another example (AV03; Pickering et al., 2019). We seriously considered your suggestion to use synthetic examples, but decided to stick with real data to pre-empt any accusation that our paper addresses a "straw man problem", as Reviewer 2 seems to suggest.

4. Additional references have been added (i) to document high $^{234}$U/$^{238}$U activity ratios in other parts of the world, (ii) to give proper credit to previous proponents of the $^{207}$Pb/$^{235}$U method, and (iii) to document Albarède (1995)'s use of matrix exponentials to solve disequilibrium problems.

5. Following Dr. Nuriel's request, we have added a back-of-the-envelope calculation showing that the absolute magnitude of disequilibrium corrections is limited to c. 4 Myr. This brings home the point that our paper is not only relevant to young samples, but also to old ones.

My co-authors and I hope that you will find our revised manuscript suitable for publication in *Geochronology*.

Sincerely yours,

Pieter Vermeesch

---

## Author Response (AR2)

Prof. Pieter Vermeesch
University College London
+44 (0)20 3108 6369
https://ucl.ac.uk/~ucfbpve/

21 July 2025

Dear Dr. Schmitt,

Thank you very much for your efficient handling of the manuscript. I have made all but one of the changes requested in your decision letter. Having made a random selection of high profile geochronology papers, I found that most of these do not define the acronym 'MSWD'. This acronym is so widespread in our field that I do not think it necessary to define it. However, if you insist then I would be happy to add a reference to McIntyre et al. (1966).

In addition to the requested changes, I have also adjusted Equation 1 slightly, making it more accurate. Finally, I have followed Dr. Mezger's advice and removed all 'subjective' or 'emotional' words from the text.

Sincerely yours,

Pieter Vermeesch